# Excellent energy storage properties in lead-free ferroelectric ceramics via heterogeneous structure design

Qizhen Chai[1], Zhaobo Liu[2], Zhongqi Deng[3], Zhanhui Peng[1], Xiaolian Chao[1], Jiangbo Lu [3] ✉, Houbing Huang [2] ✉, Shujun Zhang [4] ✉ & Zupei Yang [1] ✉

Dielectric capacitors with ultrahigh power density have emerged as promising candidates for essential energy storage components in electronic and electrical systems. They enable enhanced integration, miniaturization, and lightweight design. However, the development of dielectric materials for cutting-edge energy storage applications has been significantly limited by their low recoverable energy storage density ($W_{rec}$) and energy efficiency ($\eta$), especially at moderate electric fields. In this study, we fabricated $0.85K_{0.5}Na_{0.5}NbO_3$-$0.15Sr_{0.7}Nd_{0.2}ZrO_3$ ceramics with an outstanding energy storage performance ($W_{rec}$ ~ 7 J cm$^{-3}$, $\eta$ ~ 92% at 500 kV cm$^{-1}$; $W_{rec}$ ~ 14 J cm$^{-3}$, $\eta$ ~ 89% at 760 kV cm$^{-1}$). The exceptional energy storage performance can be primarily attributed to the heterogeneous structure, where orthorhombic and tetragonal polar nanoregions are embedded in a cubic matrix. This work provides a good paradigm for designing dielectric materials with ultrahigh energy storage density and excellent energy efficiency at a moderate applied electric field, aligning with the stringent demands for advanced energy storage applications.

Lead-free dielectric ceramics are increasingly sought after for various electrical device components due to their environmentally friendly nature, ultrahigh power density ($P_D$), ultrafast charge/discharge rate ($t_{0.9}$), and good reliability. They are particularly appealing owing to the increasing demand for enhanced energy storage density, and the need for integration and miniaturization in electronic devices[1–4]. However, these ceramics have lower recoverable energy storage density and higher energy dissipation, i.e., lower energy efficiency, compared to their lead-based counterparts. This deficiency is primarily attributed to their unsatisfactory microstructure, rendering them unsuitable for advanced electronic applications[5–7], but for environmental and health considerations, lead-based candidates are strictly limited for sustainable social development. Therefore, numerous efforts have been made to improve the performance of lead-free ceramics for energy storage dielectric capacitors[8]. Among various lead-free materials, including

$Bi_{0.5}Na_{0.5}TiO_3$ (BNT)[9], $BiFeO_3$ (BF)[10], and $BaTiO_3$ (BT)[11], $K_{0.5}Na_{0.5}NbO_3$ (KNN)-based ceramics are one of the most extensively studied dielectric for advanced energy storage applications[1–4,12]. A comprehensive analysis of the current developments in the aforementioned energy storage materials can be found in the Supporting Information. Notably, the ultrahigh $W_{rec}$ values surpassing 15 J cm$^{-3}$ have been included[13–15].

Many researchers have attempted to develop KNN-based ceramics with high $W_{rec}$ and $\eta$. Various strategies, such as high-entropy design[16], domain engineering[17,18], defect engineering[19], superparaelectric state[20,21], grain size engineering[22], and enhanced local random field[23] have been proposed to achieve high $W_{rec}$. High $W_{rec}$ values of ~10.06 J cm$^{-3}$, ~8.09 J cm$^{-3}$, and ~7.4 J cm$^{-3}$ have been realized in KNN-based bulk ceramics prepared by conventional solid-state sintering at their respective breakdown electric field of about 800 kV cm$^{-1}$[16,24,25].

[1]School of Materials Science and Engineering, Shaanxi Normal University, Xi'an, China. [2]School of Materials Science and Engineering, Beijing Institute of Technology, Beijing, China. [3]School of Physics and Information Technology, Shaanxi Normal University, Xi'an, China. [4]Institute for Superconducting and Electronic Materials, University of Wollongong, Wollongong, NSW, Australia. ✉e-mail: jblu10@snnu.edu.cn; hbhuang@bit.edu.cn; shujun@uow.edu.au; yangzp@snnu.edu.cn

While under a moderate electric field of 500 kV cm⁻¹, these materials exhibit suboptimal $W_{rec}$ values of ~5.6 J cm⁻³, ~3.8 J cm⁻³, and ~4.4 J cm⁻³, respectively, which limit their compactness and miniaturization for integration into electronic devices[1,9]. Given the practical application of electronic devices within relatively low electric field environments, a more reasonable approach is to optimize $W_{rec}$ by improving the maximum polarization ($P_{max}$), reducing the remnant polarization ($P_r$), and mitigating polarization saturation[26,27].

High $W_{rec}$ can be achieved by designing ceramics with fine grains, a dense microstructure, high polarization under external electric fields, a highly dynamic structure, an isotropic lattice structure, and minimal impurities[28–31]. Subsequently, significant challenges persist in realizing ceramics with both ultrahigh $W_{rec}$ and outstanding $η$, particularly when employing a conventional solid-state method. Introducing paraelectric end members or linear compounds into a ferroelectric ceramic is a generally accepted approach for achieving relaxor ferroelectric (RFE) ceramics. This involves disrupting the long-range ferroelectric order and creating nanodomains[32–35]. In addition, the polymorphic nanodomain structure can effectively weaken the energy barrier and polarization anisotropy, resulting in a flattened energy profile compared to single-phase nanodomains, which contributes to the simultaneous significant enhancement of $W_{rec}$ and $η$ simultaneously[8,16,36]. However, the current situation indicates that the construction of such a heterogeneous structure generally requires relatively complex chemical components and/or processes.

In this work, we propose a feasible strategy for constructing a heterogeneous nanodomain structure that encompasses orthorhombic (O, Amm2) and tetragonal (T, P4mm) phases within a cubic (C, Pm-3m) paraelectric matrix. The objective of this strategy is to address the challenges associated with achieving ceramics that possess exceptional $η$ and ultrahigh $W_{rec}$ under a moderate electric field. We developed a binary system incorporating the paraelectric end member $Sr_{0.7}Nd_{0.2}ZrO_3$ into a KNN matrix to realize RFEs, which are promising candidates for energy storage optimization due to their weakly intercoupled nanodomains[14]. Leveraging the maximum polarization potential[24], we selected KNN as the primary component. We also introduced $Sr^{2+}$ and $Zr^{4+}$ ions to modify the $T_{O-T}$ and $T_{T-C}$, resulting in the formation of an orthorhombic-tetragonal-cubic heterogeneous nanodomain structure, in line with the multiphase regulation of KNN-based ceramics[37]. Meanwhile, the rare earth ion of $Nd^{3+}$ contributes to enhancing energy storage performance by reducing grain size and optimizing the microstructure of dielectric ceramics[38,39]. As expected, we achieved a remarkable $W_{rec}$ of ~7 J cm⁻³ with an outstanding $η$ of ~92% at a moderate electric field of 500 kV cm⁻¹ and a giant $W_{rec}$ of

~14 J cm⁻³ with a high $η$ of ~89% at an electric field of 760 kV cm⁻¹, approaching $E_b$. This represents the highest level of performance achieved in KNN-based bulk ceramics, underscoring the effectiveness of the heterogeneous structure strategy in the development of advanced pulse power capacitors with exceptional $W_{rec}$ and $η$. This finding proves that a heterogeneous structure can be constructed through a multi-scale process (Fig. S1, Supporting Information) using simple chemical compositions and conventional solid-phase reactions. These insights are invaluable for the development of high-performance energy storage materials.

## Results

### Energy storage performance and charge/discharge properties for practical application

Unipolar $P$-$E$ hysteresis loops were measured under 200 kV cm⁻¹ for $(1-x)$KNN-$x$SNZ ceramics. The corresponding $P_{max}$ and $P_r$ values, as well as energy storage properties ($W_{rec}$, $W_{total}$, and $η$ were calculated using $W_{rec} = \int_{P_r}^{P_{max}} EdP$, $W_{total} = \int_0^{P_{max}} EdP$, and $η = \frac{W_{rec}}{W_{total}} \times 100\%$, respectively), are presented in Fig. S2a–c, (Supporting Information). Comprehensively, ceramics with $x = 0.15$ exhibit a relatively strong energy storage capacity, with $W_{rec}$ reaching ~1.6 J cm⁻³ and $η$ approaching 91% (Fig. S2c, Supporting Information). Additionally, a high $W_{rec}$ of ~1.6 J cm⁻³ with a large $η$ of ~86% and a large $W_{rec}$ of ~1.5 J cm⁻³ with an ultrahigh $η$ ~ 93% can be obtained in samples with $x = 0.14$ and $x = 0.16$, respectively. This is attributed to their slender $P$-$E$ shapes (Fig. S2a, Supporting Information) and significantly reduced $P_r$ (Fig. S2b, Supporting Information). Furthermore, the unipolar $P$-$E$ hysteresis loops of the ceramics were measured up to their respective breakdown electric fields. The results are illustrated in Fig. 1a–d ($x ≥ 0.14$) and S3a–d (Supporting Information, $x < 0.14$). The slender unipolar $P$-$E$ loops of KNN-$x$SNZ ceramics ($x = 0.14$, 0.15, 0.16, and 0.18; see Fig. 1a–d) under various electric fields showcase the canonical relaxor behaviors. To further explore the properties of the ceramics, the energy storage parameters ($W_{rec}$, $W_{total}$, and $η$) are depicted in Fig. 1e–h and S3e–h (Supporting Information). Owing to the increasing polarization of ceramics, $η$ generally improves under low electric fields. However, under high electric fields, the dielectric loss and leakage current increase because of the increased dielectric heating and nonlinear effect of the dielectrics, meanwhile the field induces defects, thus increasing leakage current, negatively affecting $η$ (Fig. S4, Supporting Information). Even though the efficiency slightly decreases near the breakdown strength, it can be further enhanced by optimization of grain size and phase distribution. A $W_{rec} < 5$ J cm⁻³ can be observed in a composition range of $x < 0.14$,

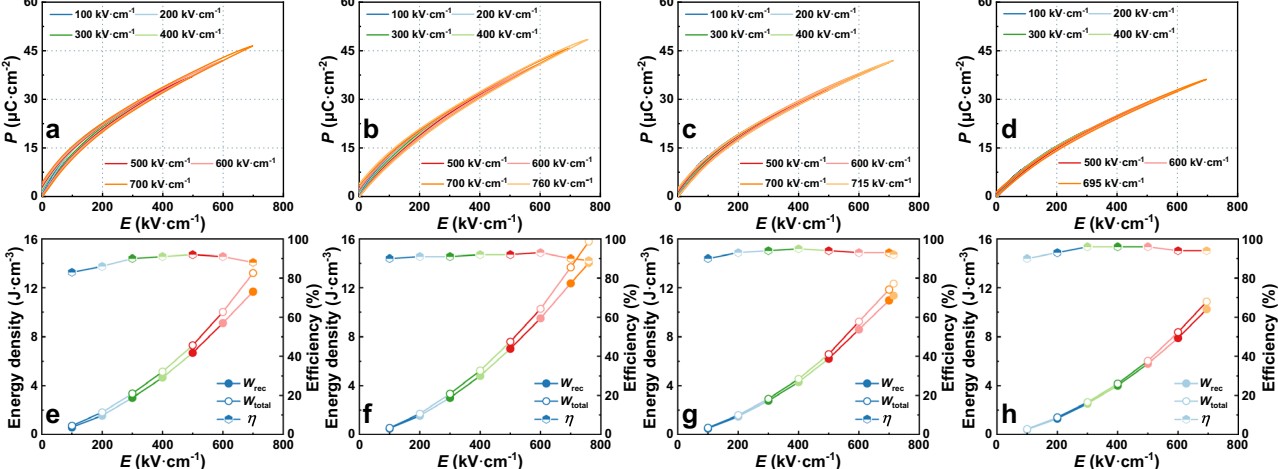

**Fig. 1 | Energy storage properties of $(1-x)$KNN-$x$SNZ ceramics. a–d** Ambient temperature $P$-$E$ loops measured under various electric fields of **a** $x = 0.14$, **b** $x = 0.15$, **c** $x = 0.16$, **d** $x = 0.18$. **e–h** Composition-dependent energy storage parameters ($W_{rec}$, $W_{total}$, and $η$) of **e** $x = 0.14$, **f** $x = 0.15$, **g** $x = 0.16$, **h** $x = 0.18$.

accompanied by an $\eta < 80\%$. In contrast, ceramics with $x \geq 0.14$ consistently maintain superior energy storage performance with $W_{rec}$ beyond 10 J cm$^{-3}$ and simultaneously high efficiency ~90% under electric fields approaching their $E_b$ values. Of particular significance is that a superior energy storage density of 14 J cm$^{-3}$ with a high $\eta$ of 89% have been achieved in composition of $x = 0.15$ ceramic under an electric field of 760 kV cm$^{-1}$. Meanwhile, a record-high $\eta$ of 94% is achieved in $x = 0.18$ ceramic with a large $W_{rec}$ of 10 J cm$^{-3}$ under an electric field of 695 kV cm$^{-1}$. Fig. S5 (Supporting Information) shows the corresponding $P_{max}$, $P_r$ and $\Delta P$ ($P_{max}$ - $P_r$) values of ceramics with $x \geq 0.14$ obtained from the $P$-$E$ loops (Fig. 1a–d), indicating the appropriate introduction of SNZ can maintain a large $P_{max}$ of 48.5 μC cm$^{-2}$ while reducing $P_r$ to 1.4 μC cm$^{-2}$ prior to critical breakdown electric fields. It should be noted that the ceramics with $x \geq 0.14$ in this study exhibit excellent overall energy storage properties and wide-composition stability characteristics. This can be attributed mainly to the high $\Delta P$ and the delayed polarization saturation (compare Fig. 1a–d to S3a–d, Supporting Information), demonstrating a promising candidate for practical application in advanced energy storage capacitors. Additionally, for $x \geq 0.15$, both $E_b$ and $P_{max}$ decrease with increasing SNZ content, indicating that ceramics with $x = 0.15$ exhibit the best properties. Considering the $W_{rec}$ beyond 10 J cm$^{-3}$, we conclude the optimal doping ratio range of SNZ within KNN for composition range is 0.14–0.18.

As summarized in Fig. 2a, this study signifies a significant breakthrough in KNN-based ceramics, surpassing previously documented results[2,16,24–26,36,40–42]. Furthermore, comparisons of $W_{rec}$ values under different electric fields between the $x = 0.15$ ceramics and other actively studied KNN-based dielectric ceramics are recorded in Fig. 2b. As seen, a large $W_{rec}$ of 7 J cm$^{-3}$ with high $\eta$ of 92% is obtained at an electric field of 500 kV cm$^{-1}$ for $x = 0.15$ ceramics, demonstrating superior comprehensive energy storage performance under moderate electric fields. This further indicates that the high $\Delta P$ plays a more crucial role in the overall energy storage properties. Another critical aspect of evaluating advanced dielectric capacitors for real applications is their charge/discharge performance. The underdamped discharge behaviors of the $x = 0.15$ ceramic under various electric fields are illustrated in Fig. 2c, d. These behaviors are followed by a linear increase in positive current peak ($I_{max}$), current density ($C_D$, calculated as $C_D = I_{max}/S$), and power density ($P_D$, calculated as $P_D = EI_{max}/S$) up to

32 A, 4120 A cm$^{-2}$, and 930 MW cm$^{-3}$, respectively, under 450 kV cm$^{-1}$. Furthermore, overdamped discharge properties are presented in Fig. 2e–g, showing a high discharge energy density ($W_d$, calculated as $W_d = R \int i(t)^2 dt/V$) of ~5.3 J cm$^{-3}$ and an ultrafast discharge rate ($t_{0.9}$) of ~39 ns at 450 kV cm$^{-1}$ (Fig. 2f). Figure 2h shows that the $W_d$ of most lead-free systems is below 4 J cm$^{-3}$ [2,4,9–12,14–16,18,22,25,27,33,35,40,42–44], while the studied 0.85KNN–0.15SNZ ceramic exhibits a high test electric field and $W_d$ value. In addition, the superior stability (Fig. S6, Supporting Information) of this material enhances its practical application in demanding energy storage conditions. All these findings suggest that the 0.85KNN–0.15SNZ ceramics hold significant promise for use in advanced capacitors.

## Elucidation of O-T-C multiphase nanoregion coexistence by a multi-scale process

To explore the underlying mechanism of the improved energy storage properties, we investigated the phase structure of the current system (Fig. 3 and S7, S8, Supporting Information). Figure 3a presents the Rietveld refinement result of the 0.85KNN-0.15SNZ ceramic over a range of 20° to 80°. It reveals the coexistence of the orthorhombic (O, Amm2) and tetragonal (T, P4mm) phases, with weight fractions of 26.14% and 73.86%, respectively. The absence of rhombohedral phases, which can appear under certain composition or temperature conditions, is due to high-temperature sintering that stabilizes the T and O phases. Notably, no secondary phase is observed, indicating the complete incorporation of the SNZ into the KNN lattice (Fig. S7a, Supporting Information), which can be attributed to optimized synthesis processes and rational composition design. In addition, the enlarged view depicted in Fig. S7b, Supporting Information confirms that as $x$ increases, the phase gradually transitions into a low-polar pseudo-cubic phase structure, which is conducive to achieving low $P_r$ and high $\Delta P$. Additional insights into the structural characteristics of 0.85KNN-0.15SNZ ceramics across the temperature range of 25–600 °C were obtained through temperature-dependent powder X-ray diffractometer (XRD) techniques, as depicted in Fig. 3b and S7d (Supporting Information). These images (Fig. 3b) display amplified diffraction peaks of (110), (200), and (210) corresponding to the pseudo-cubic phase structures. The relative intensity and the number of characteristic peaks remain constant over the studied temperature range, except for a shift towards lower angles associated with lattice

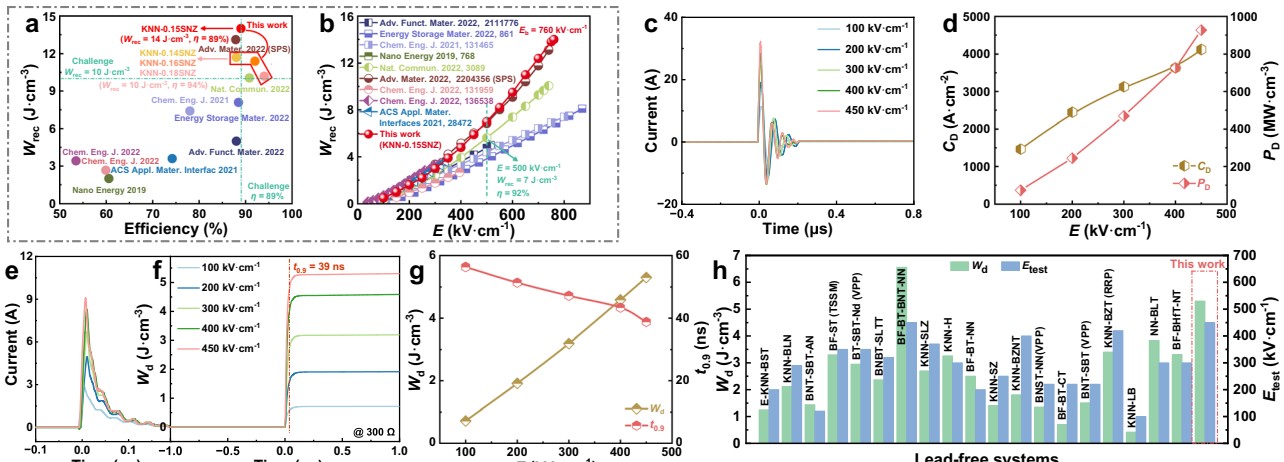

**Fig. 2 | Energy storage and charge/discharge performance of (1-$x$)KNN-$x$SNZ ceramics. a** Comparisons of the energy storage properties between the studied ceramics ($x \geq 0.14$) in this work and other recently reported KNN-based ceramics. **b** Comparisons of the $W_{rec}$ between the $x = 0.15$ ceramic and other recently reported KNN-based ceramics at various electric fields. Charge/discharge performance of $x = 0.15$ ceramic: **c** underdamped discharge waveforms and **d** corresponding $C_D$ and $P_D$ values at various electric fields; **e** overdamped discharge waveforms, **f** discharge energy density, and **g** corresponding $W_d$ and $t_{0.9}$ values at various electric fields ($R = 300\,\Omega$). **h** Comparisons of the $W_d$ and test electric field between the $x = 0.15$ ceramic in this study and recent publications (bulk ceramics as well as thick films).

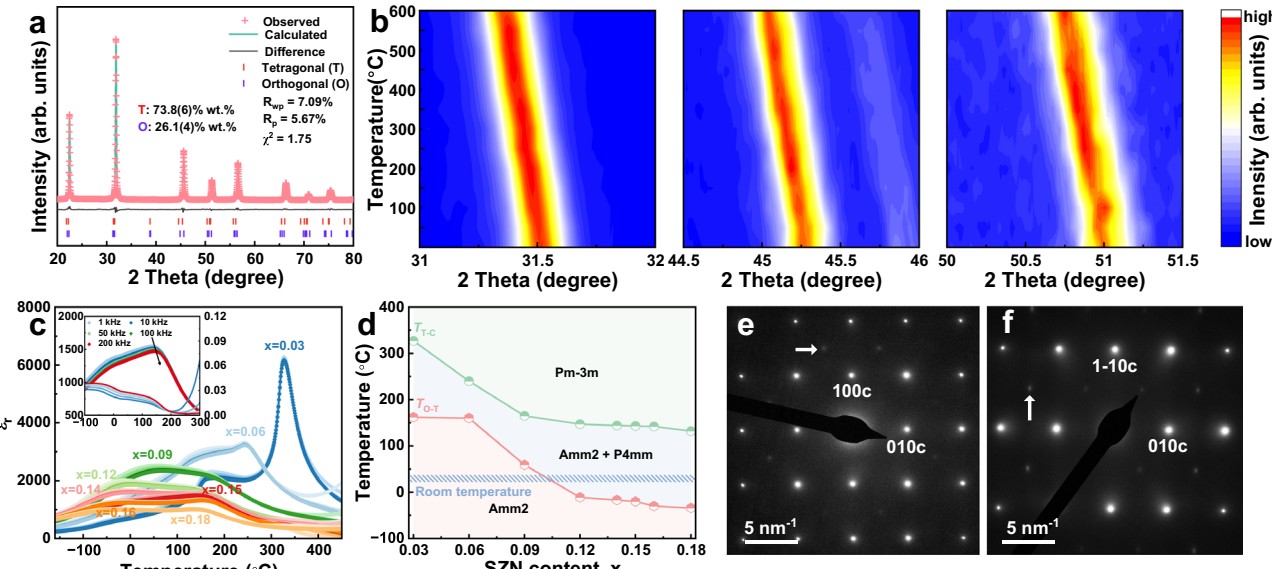

**Fig. 3 | Phase structures and dielectric properties of (1-x)KNN-xSNZ ceramics.**
**a** Rietveld refinement XRD pattern of $x = 0.15$ ceramic. **b** Temperature-dependent XRD patterns of $x = 0.15$ ceramic collected in the temperature range of 25 °C to 600 °C. **c** Temperature-dependent dielectric constant at various frequencies. Inset presents the dielectric constant and dielectric loss of $x = 0.15$ ceramic as a function of temperature, measured at various frequencies ranging from 1 kHz to 200 kHz. **d** Phase diagram of (1-x)KNN-xSNZ ceramic. **e** Selected area electron diffraction patterns along the [001]c, and **f** along [110]c of $x = 0.15$ ceramic.

expansion, indicating the absence of any phase structure changes[8]. On the other hand, as shown in Fig. S7c, d (Supporting Information), the phase structure gradually changes from O (25 °C) phase to T (-200 °C) phase and then to C (-350 °C) phase for $x = 0.03$ ceramic, in contrast to $x = 0.15$ ceramic. The crystal phase feature of the $x = 0.15$ ceramic is primarily attributed to the coexistence of O-T phases, in other words, the pseudo-cubic phase structure. The temperature-dependent dielectric constant at different frequencies of all components (Fig. 3c) is consistent with the above analysis. Three main characteristics can be derived from the dielectric properties: 1) The presence of an extremely broad and flat dielectric peak with frequency dispersion behavior ($x \geq 0.06$, see Fig. 3c), which can be further verified by the inset in Fig. 3c for $x = 0.15$ sample, represents a canonical relaxor feature attributed to the presence of PNRs[8]. 2) The relatively high room temperature dielectric constant ($\geq 1000$) observed in the current systems ($x \geq 0.06$) is conducive to achieving high $P_{max}$. 3) It can be noted that all samples exhibit two dielectric anomaly peaks (temperature denoted by "$T_{O-T}$" and "$T_{T-C}$"), which represent the transitions from O to T phase and from T to C phase, respectively, upon heating[12]. The $T_{O-T}$ and $T_{T-C}$ data points of (1-x)KNN-xSNZ ceramics at a frequency of 1 kHz are extracted to construct a phase diagram to identify the composition dependence of the phase structures, as depicted in Fig. 3d. It can be observed that both $T_{O-T}$ and $T_{T-C}$ decrease with increasing x, leading to the formation of a coexisting O-T phase in the composition of $x \geq 0.12$.

Additionally, the Raman spectra presented in Fig. S8 (Supporting Information) not only confirm the enhanced relaxor behavior but also support the conclusion that the phase structure of (1-x)KNN-xSNZ samples transforms from the O phase to pseudo-cubic phase as x increases[37]. Furthermore, the selected area electron diffraction patterns taken along [001]c and [110]c directions indicate that the sample exhibits an average cubic structure, as illustrated in Fig. 3e, f, respectively. The diffused and weak spots as indicated by white arrows confirm the existence of orthorhombic structure "$\sqrt{2}$ unit cell" which is orientated along the cubic [110]c in the x-direction, the [1–10]c in the y-direction and the [001]c in the z-direction[16].

In addition, Fig. S9 (Supporting Information) shows a significant refinement in the average grain size (AGS) calculated by Nano

Measurer for the ceramics, particularly for $x \geq 0.06$, with the smallest grain size observed for $x = 0.12$–0.15, with values being on the order of 150 nm. The findings from field emission scanning electron microscope (FE-SEM) analysis support a uniform and compact microstructure with low apparent porosity and refined grains for the $x = 0.15$ ceramic, as further illustrated in Fig. S10a (Supporting Information). These characteristics have a favorable impact on $E_b$. Meanwhile, due to the confinement imposed by grain boundary conditions, the size of domains decreases as grain size decreases. In other words, the refined grain boundaries can effectively hinder domain growth, which is an important factor for achieving PNRs.

We then applied the vertical piezoresponse force microscopy technique to uncover the domain morphology under various DC tip bias voltages. Figure 4a₁–h₁ and a₂–h₂, respectively, illustrate the effects of applied voltages on the dynamic behavior of domain switching in the pure KNN and $x = 0.15$ ceramics at ambient temperature. The amplitude diagrams record the piezoresponse strength, while the phase diagrams represent the polarization orientation[18,45–47]. Applying a driving voltage of merely 3 V (Fig. 4a, e) leads to the emergence of striped ferroelectric domains in pure KNN ceramic, while nanoscale domains are observed in the $x = 0.15$ ceramic. Fig. S10b (Supporting Information) further shows the striped ferroelectric domains in pure KNN ceramic. As the DC tip bias voltage increases to 3 V, partial reversal of the domains is observed (Fig. 4b, f). For $x = 0.15$ ceramic, a voltage of 12 V is adequate to induce complete reversal, with the polarized domains easily reverting when a -12 V tip bias is applied (Fig. 4c₂, d₂, g₂, h₂). In stark contrast to the $x = 0.15$ ceramic, the striped ferroelectric domains remain clearly distinguishable for the pure KNN ceramic (Fig. 4c₁, d₁, g₁, h₁), suggesting that the $x = 0.15$ ceramic gives rise to the emergence of highly dynamic polar nanoregions (PNRs). PNRs, characterized by local structural heterogeneity, confirm enhanced relaxor behavior and contribute to slimmer P-E loops due to their high sensitivity to external electric fields[40,47]. The topography of piezoresponse force microscopy (PFM) image presented in Fig. 4i underscores the homogeneous and dense microstructure, as well as small AGS on the order of 150 nm. Figure 4j displays the switching spectral piezoelectric response force microscopy (SSPFM) loops at ambient temperature for detecting the local piezoresponse of the

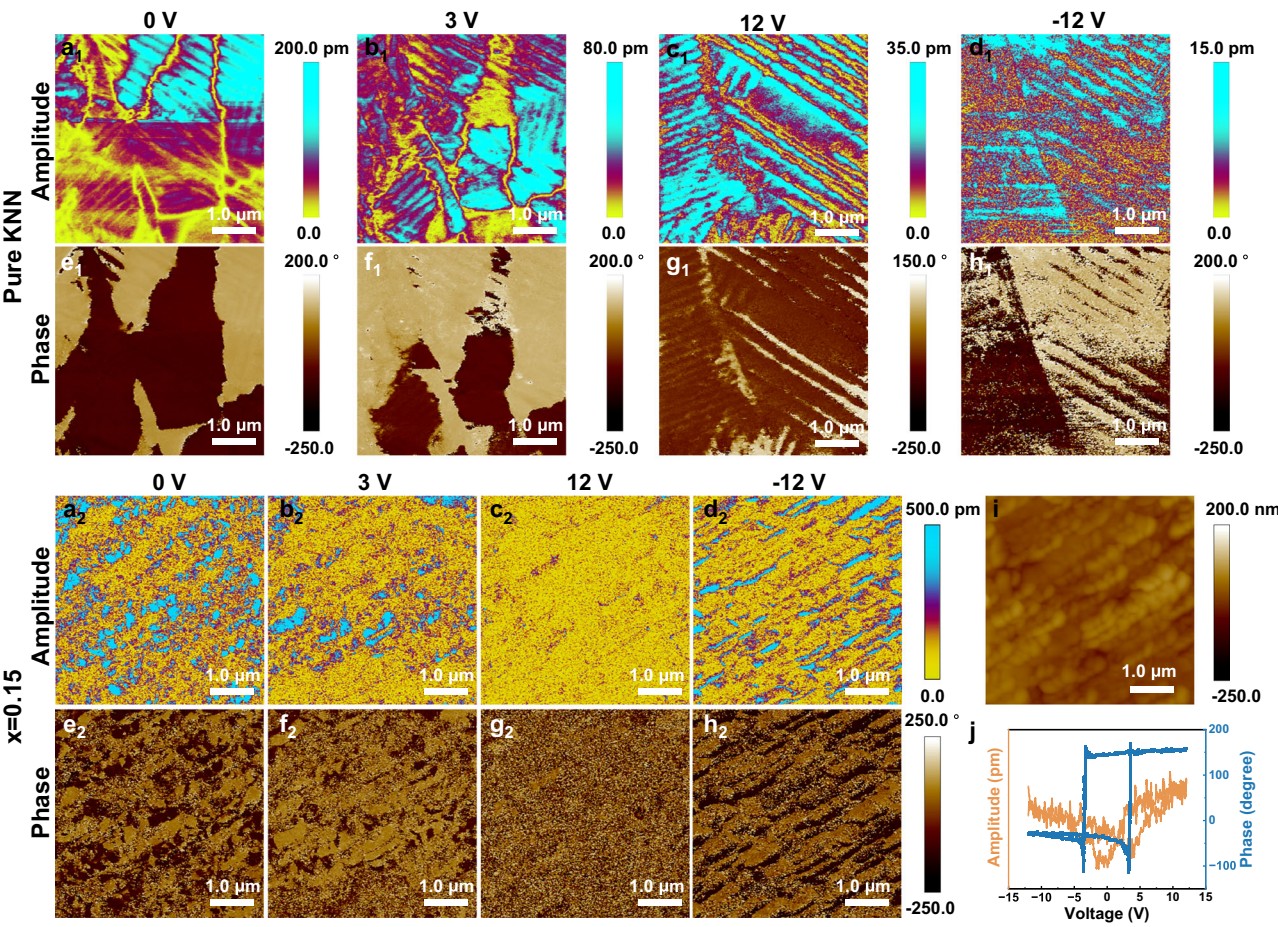

**Fig. 4 | Out-of-plane PFM of (1-x)KNN-xSNZ ceramics.** $a_1$–$h_2$ Amplitude and phase images at different voltages of $a_1$–$h_1$ pure KNN, and $a_2$–$h_2$ $x = 0.15$ ceramic. **i** PFM topography, and **j** SSPFM hysteresis loops of $x = 0.15$ ceramic.

0.85KNN–0.15SNZ ceramic. The amplitude and phase hysteresis loops are recorded by applying ±12 V DC voltage on the selected domain region. Compared to previous studies[48,49], the 0.85KNN-0.15SNZ ceramic exhibits a high phase angle and a well-defined saturated square phase loop with a phase angle close to 180°, indicating the appearance of two reverse-parallel reversible polarization states under an applied electric field. The nearly perfect rectangular phase hysteresis loop provides direct evidence for a well-defined polarization along the direction of the applied electric field[50]. The amplitude hysteresis loop of the same domain region, as illustrated by Fig. 4j, exhibits a typical, asymmetric, and saturated butterfly loop. The asymmetry can be explained by the presence of a built-in internal field and domain movement under the tip region[51]. The origin of a built-in internal field is inferred to be the substitution of $Sr^{2+}$ and $Nd^{3+}$ at the A-site, leading to the generation of vacancies and distortion of the oxygen octahedron. This results in a more facile displacement of ions at the B-site and enhanced polarization. In addition, the formation of dipoles between $Sr^{2+}/Nd^{3+}$ and vacancies at non-equilibrium positions creates a built-in internal field, which enhances the relaxation behavior of ceramics[52].

We now shift our focus to the intrinsic origins of the enhanced energy storage properties. Atomic resolution high-angle annular dark-field (HAADF) scanning transmission electron microscopy was performed to further analyze the local polarization configuration of PNRs. Figure 5a shows the atomic resolution HAADF-STEM image taken along [001]c direction, where the integrated intensity is a function of average atomic number Z of atomic columns scanned by the electron probe[53], B sites atoms are the brighter and larger while A sites atoms are darker and smaller. Figure 5b presents the corresponding local polarization configuration of Fig. 5a derived from the 2D Gaussian peak fitting and

quantification[54]. The T phase can be obviously determined by the arrows along the horizontal and vertical directions. The nearly C phase can be confirmed by the arrows with near-zero polarization magnitude whereas the O phase can be determined by the arrows oriented close to the diagonal direction (i.e., <110>c) of Fig. 5b. The alternating polar and non-polar regions can be clearly distinguished, and the size of nano-polar region is determined to be around 3 nm. Figure 5c presents the statistic histogram of polarization magnitude of Fig. 5b. The polarization magnitudes are in the range of 0 and 9 pm, and the averaged polarization magnitude is only ≈3.42 pm, which is consistent with X-ray diffraction refinement data indicating the averaged phase is pseudo-cubic, as illustrated in Fig. 3a and S7b (Supporting Information). The atomic resolution HAADF-STEM image taken along [110]c is given in Fig. S10c (Supporting Information). The coexistence of O, T, and C multiphase PNRs exhibits a rapid response to applied electric fields, which is of great significance for the reduction of $P_r$. Moreover, for the $x = 0.15$ sample, the TEM micrograph and corresponding elemental distribution are presented in Fig. 5d. Evidently, the 0.85KNN–0.15SNZ ceramic exhibits refined grain size and highly uniform distributions of K, Na, Nb, Sr, Nd, Zr, and O, all of which contribute to achieving high $E_b$.

## Phase-field simulations of domain structures

Theoretical phase-field simulations were performed to simulate the binary (1-x)KNN-xSNZ ($x = 0.12$, 0.15, and 0.18) solid solutions[8,17]. We found that the studied compositions with $x = 0.12$, $x = 0.15$, and $x = 0.18$ indeed possess a disordered structure, where O and T nanodomains are embedded in a cubic matrix (Fig. 6 and Fig. S11, Supporting Information). The randomly distributed nanodomains with different

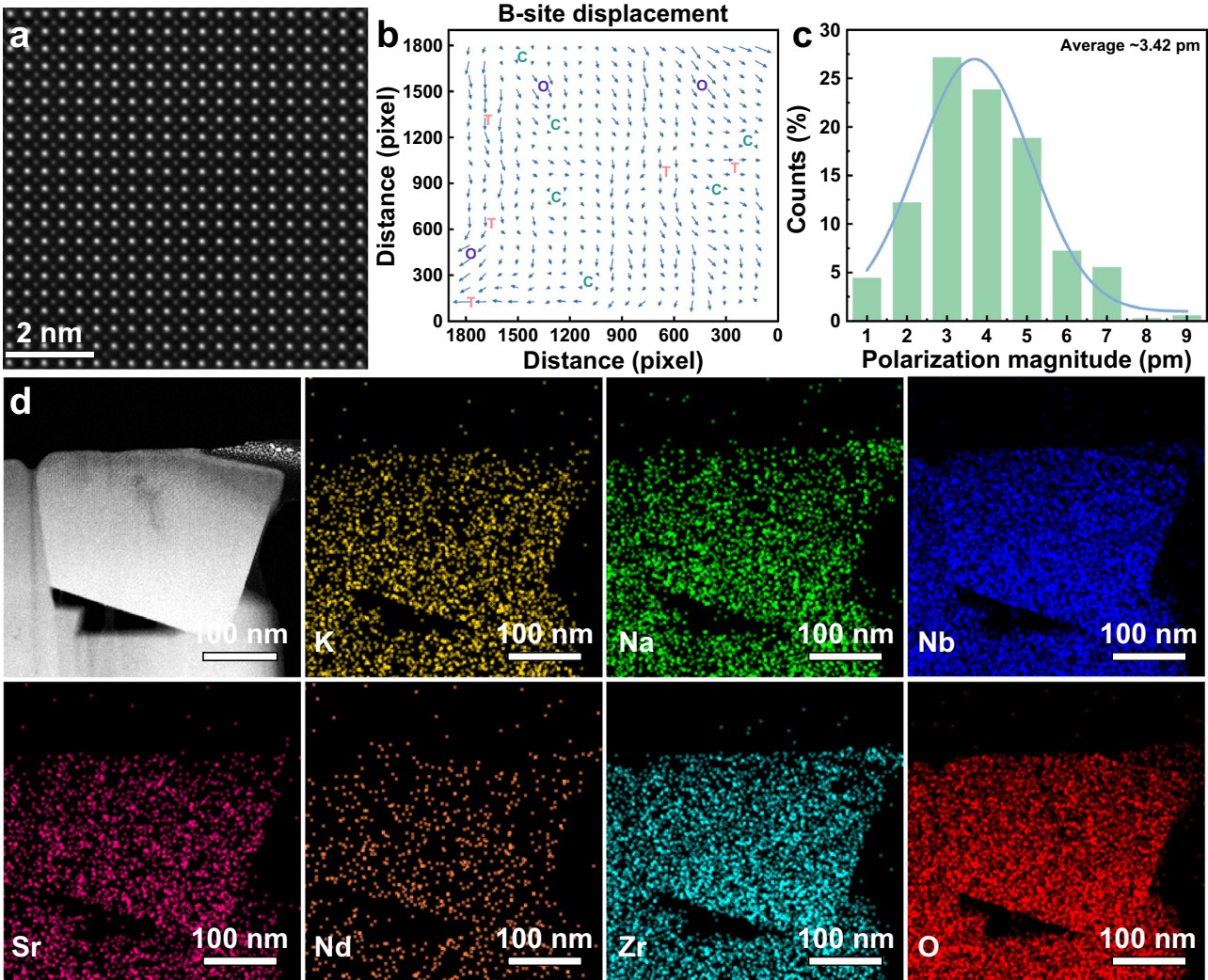

**Fig. 5 | Structural characterizations of the $x$ = 0.15 ceramic. a** Atomic resolution HAADF-STEM image along [001]$_c$ direction. **b** Quantified local polarization configuration corresponding to **a**. **c** Statistic histogram of polarization magnitude of **b**. **d** A bright-field STEM image and corresponding chemical EDS elemental mapping of 0.85KNN−0.15SNZ ceramic.

symmetries and polarization magnitudes make a significant contribution to reduced energy loss and delayed saturation. Moreover, we found the fraction of the O nanodomains decreases evidently with $x$ increase, being responsible for the relatively low $P_r$. Apart from multiphase nanodomains, a significant refinement of the domain size can be observed by increasing $x$, facilitating polarization rotation and enhancing the relaxation behavior, subsequently contributing to the reduced hysteresis and enhanced $\eta$. However, the local polarization is weakened due to a decrease in the polar phase ratio, which ultimately compromises $P_{max}$ and, hence, $W_{rec}$ for larger $x$ contents. These simulation results are consistent with our experimental findings, highlighting the benefits of the heterogeneous structure design in hysteresis and polarization, signifying an impressive advancement in energy storage performance. We also created a comparative table (Table S5, Supporting Information) that distinguishes the contributions of different mechanisms, while in the current research, the formation of heterogeneous structure is the dominant mechanism responsible for the observed energy storage properties.

### Optical transparency of the relaxor ceramics

In addition to the excellent energy storage performance, (1-$x$)KNN-$x$SNZ samples exhibit satisfactory transparency, which may find

application in some specific scenarios, such as energy storage components in solid-state lighting, electro-optical devices, and even biomedical materials. Fig. S12 (Supporting Information) gives the optical transmittance ($T$%) and physical picture. It can be found that as the SNZ content ($x$) increased from 0.03 to 0.18, the $T$% values in the visible regions exhibited an initial increase followed by a decrease. The highest value of 64% was observed at $x$ = 0.09, as evidenced by the physical sample. Notably, the ceramics with $x$ = 0.06, $x$ = 0.12, and $x$ = 0.15 demonstrate exceptional $T$% values up to 78% at a wavelength of 2000 nm, indicating outstanding transparency at the near-infrared region.

### Discussion

In summary, we propose an innovative design strategy for lead-free relaxors, characterized by a heterogeneous structure that is constructed through a multi-scale process from refining grain boundaries to domains, and then to O-T-C multiphase nanoclusters. Notably, the 0.85KNN-0.15SNZ bulk ceramics achieve exceptional energy storage capabilities ($W_{rec}$ ~ 14 J cm$^{-3}$, $\eta$ ~ 89%). This is further realized by 0.82KNN-0.18SNZ bulk ceramics possessing an ultrahigh $\eta$ of 94% with a remarkable $W_{rec}$ of 10 J cm$^{-3}$. Based on such excellent energy storage properties and wide-composition stability characteristics, the findings in this study not only introduce highly competitive candidates for

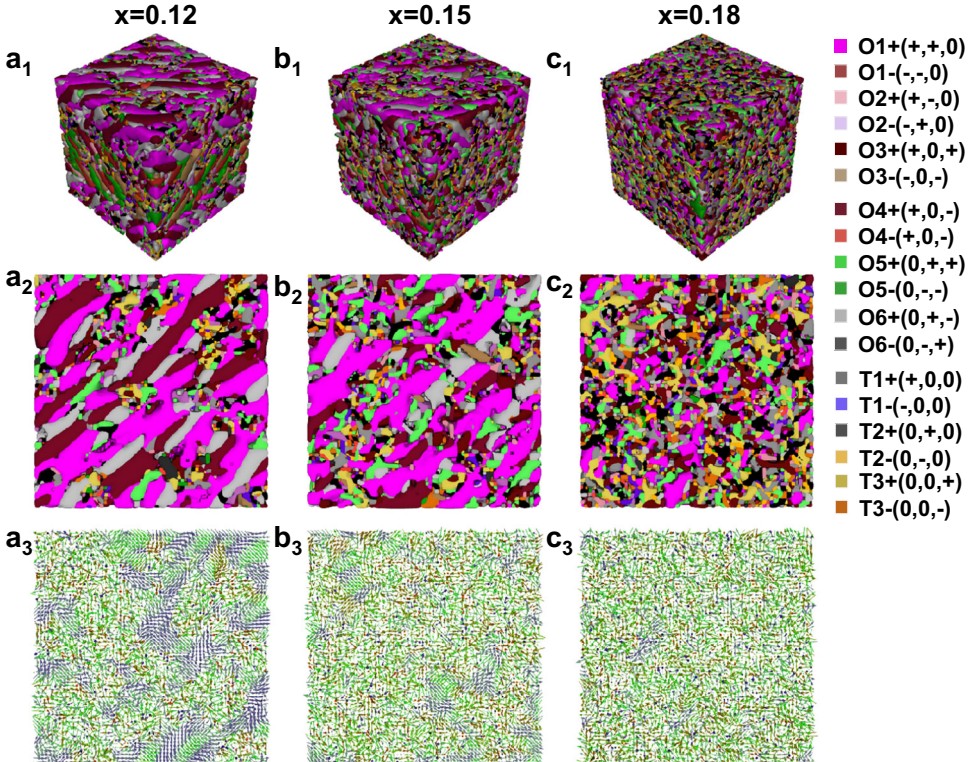

**x=0.12**        **x=0.15**        **x=0.18**

Legend:
- O1+(+,+,0)
- O1-(-,-,0)
- O2+(+,-,0)
- O2-(-,+,0)
- O3+(+,0,+)
- O3-(-,0,-)
- O4+(+,0,-)
- O4-(+,0,-)
- O5+(0,+,+)
- O5-(0,-,-)
- O6+(0,+,-)
- O6-(0,-,+)
- T1+(+,0,0)
- T1-(-,0,0)
- T2+(0,+,0)
- T2-(0,-,0)
- T3+(0,0,+)
- T3-(0,0,-)

**Fig. 6 | Phase-field simulations of the domain structures. $a_1$–$c_1$** Three-dimensional domain structures of **$a_1$** $x = 0.12$, **$b_1$** $x = 0.15$, **$c_1$** $x = 0.18$ with coexisting O and T nanodomains (the cubic matrix is omitted for clarity). Two-dimensional domain structures of **$a_2$** $x = 0.12$, **$b_2$** $x = 0.15$, **$c_2$** $x = 0.18$. Sectional view of the three-dimensional domain structures of **$a_3$** $x = 0.12$, **$b_3$** $x = 0.15$, **$c_3$** $x = 0.18$. The various colors indicate different polarization orientations of the nanodomains.

advanced pulsed power applications but also open new avenues for designing high-performance materials in diverse areas.

## Methods

### Ceramics preparation

The $(1-x)K_{0.5}Na_{0.5}NbO_3$-$xSr_{0.7}Nd_{0.2}ZrO_3$ ($(1-x)$KNN-$x$SNZ) ceramics were fabricated through a conventional high-temperature solid-state sintering method[12,18,26,30]. Raw powders, including $K_2CO_3$ (≥99.99%), $Na_2CO_3$ (≥99.999%), $Nb_2O_5$ (≥99.99%), $SrCO_3$ (≥99.99%), $Nd_2O_3$ (≥99.99%), and $ZrO_2$ (≥99.99%) from Aladdin, were utilized. Stoichiometric quantities of these raw materials were weighed and subjected to 24 h of ball milling in ethanol using yttrium-stabilized zirconia grinding media. Subsequently, the mixture underwent calcination at 850 °C for 5 h before a second 24 h ball milling. Following this, the dried powders were cold isostatically pressed into pellets with a diameter of 11.5 mm and sintered at temperatures ranging from 1200 to 1300 °C for 5 h. All the samples were polished down to a thickness of 0.1 mm and electrode with fired-on silver paste with a diameter of 1 mm, for the energy storage properties evaluation.

### Electrical property measurements

Ferroelectric measurements: the polarization-electric field (*P-E*) hysteresis loops were recorded with a ferroelectric test system (Premier II, Radiant, USA). Charge/discharge measurements: charge/discharge capabilities were evaluated with a charge/discharge instrument (CFD001, Gogo Instruments Technology, China). Dielectric measurements: dielectric properties were characterized at different temperatures and frequencies using an LCR meter (Agilent E4980A).

### Optical performance measurements

Optical transparency: transmittance was estimated using UV-vis-NIR spectroscopy (PerkinElmer Lambda 1050).

### Structure characterizations

Phase structure measurements: the phase structures were examined using a high-resolution XRD (SmartLab-9, Rigaku, Japan) (Rietveld structure refinement was performed using the GSAS program) and a Raman spectrometer (inVia Reflex, Renishaw, UK). Scanning electron microscopy: Microstructural morphology was recorded utilizing an FE-SEM (SU8020, Japan). Domain structure: The observation of domain morphology was conducted through PFM (Dimension ICON, US Bruker Nano Inc, USA) and field emission transmission electron microscope (Talos F200i, Thermo-Fisher Scientific Inc, USA). Transmission electron microscopy: Atomic resolution HAADF scanning transmission electron microscopy investigations and selected area electron diffraction analysis were performed on a ThermoFisher Scientific transmission electron microscope Titan Cubed Themis G2 300 microscope with a probe aberration corrector operated at 300 kV. Specimens for TEM observations were prepared by the conventional method, including cutting and mechanical polishing, and finalized by Ar+ ion beam thinning and polishing with Gatan Precision Ion Polishing System 695.

### Phase-field simulations

The method of phase-field simulation is elaborated in detail in the ESI.

## Data availability

All data supporting this study and its findings are available within the article and its Supplementary Information. Any data deemed relevant are available from the corresponding author upon request.

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

## Acknowledgements
This work was supported by the National Natural Science Foundation of China (no. 52272119, Z.P.Y.).

## Author contributions
Conception and design were performed by Q.Z.C., Z.H.P., and S.J.Z. Sample fabrication, testing (including energy storage, dielectric, optical, structural, and other properties), and processing were performed by Q.Z.C. STEM and TEM measurements and analysis, were performed by Z.Q.D. and J.B.L. Phase-field simulations were performed by Z.B.L. and H.B.H. Data management and organization involved contributions from all authors. The manuscript was drafted by Q.Z.C., with supervision and revisions provided by X.L.C., Z.P.Y., and S.J.Z. Z.P.Y. served as the primary contact for the submission of the paper.

## Competing interests
The authors declare no competing interests.
