## [Transparent Peer Review file · Nature Communications]

Excellent energy storage properties in lead-free ferroelectric ceramics via heterogeneous structure design

Corresponding Author: Professor Zupei Yang

Version 0:

Reviewer comments:

Reviewer #1

(Remarks to the Author)

This work presents a heterojunction design strategy aimed at further improving energy storage performance, particularly at relatively low electric fields. Overall, the research is systematic and of high quality, and I would not be surprised to see it published in other high-impact journals. However, to meet the standards for Nature Communications, the authors need to address the following points, particularly the first one, before acceptance, as part of a major revision:

1. SNZ Dopant Identification and Heterojunction Design: How is $\text{Sr}_{0.7}\text{Nd}_{0.2}\text{ZrO}_3$ (SNZ) identified as a dopant in the KNN lattice? Is the improvement in this work solely attributed to the heterojunction design? For instance, adding external dopants to form a heterostructure can also involve other mechanisms such as high-entropy, domain, defect, or grain size engineering. Please clarify how these mechanisms are distinguished and discuss how the current design fits within or diverges from them. It is critical to highlight the key differences, novelty, and uniqueness of the approach to meet the acceptance standards of Nature Communications.
 2. Breakdown Voltage or Field: What is the breakdown voltage or field of this material? Please provide specific values, as this is crucial for understanding the material's practical application limits and its performance under high electric fields.
 3. Energy Efficiency vs. Electric Field: Please include a detailed discussion on the relationship between energy efficiency and electric field. How does the electric field affect energy efficiency, and what factors contribute to the observed changes? A graph or a more explicit relationship between these parameters would be beneficial to the readers.
 4. SNZ Doping Ratio of Up to 0.18: How was the optimal doping ratio range of ~ 0.18 for SNZ in the KNN lattice identified? Please provide more details on the experimental or theoretical justification behind this choice, along with a discussion of the effects of varying the doping ratio on the material's energy storage performance and efficiency.
 5. Efficiency Drop at Higher Electric Fields (92% to 89%): Why does the energy efficiency drop from 92% to 89% at higher electric fields? Please discuss the potential causes, such as increased dielectric losses or structural limitations at higher fields. Additionally, suggest possible mitigation strategies, such as optimizing grain size or phase distribution, to maintain high efficiency even at elevated fields.
 6. Phase Composition (O-T-C) and Absence of Secondary Phases: Is there any evidence of rhombohedral phases, or is the material strictly limited to orthorhombic, tetragonal, and cubic (O-T-C) phases? Also, why are no secondary phases observed? This point could be clarified by further discussing the synthesis process and the stability of the material under varying conditions.
 7. Material System Reliability and Stress Testing: What is the reliability of this material system over time? To assess its practical use in real-world applications, please consider adding stress tests to check whether energy storage properties degrade under mechanical, thermal, or electrical stress. This will provide insights into the material's long-term performance and robustness.
 8. SNZ Ratio and its Effect on P_{max} and P_{r} in Figure S1(b): In Figure S1(b), why does the SNZ ratio initially cause the maximum polarization (P_{max}) and remnant polarization (P_{r}) to increase but decrease after a certain point? Please explain the underlying mechanism, especially around the observed maximum, and discuss the correlation between the SNZ ratio and energy efficiency.
 9. Scalability: Will larger areas increase leakage and affect energy storage performance? Is the randomness in the heterogeneous structure sufficient to cover leakage variability, or is there still an area dependency that impacts normalized energy storage efficiency? Addressing this would clarify the material's suitability for large-scale applications.
- (PDF version also attached)

Reviewer #2

(Remarks to the Author)

(A .pdf file is attached for better format)

In this manuscript, the authors realized KNN-SNZ based, lead free dielectric capacitors with outstanding energy storage performance. For explanations of the enhanced performance, multi-phase coexistence is proposed and supported by both experiment and simulation.

The manuscript is well organized, and the evidence for the explanation is to some degree comprehensive. On the other hand, there are weak points in the manuscript, especially the inconsistency of the evidence across the figures.

To improve the overall quality, the following questions and comments need to be addressed:

1. Please include the importance of being lead-free in this material system. The manuscript only mentions the comparison in performance perspective but doesn't convince readers why being lead-free is important.
2. What are the size and thickness of the devices? Does the performance of the devices depend on the device size?
3. The record-high W_{rec} of 14 J cm^{-3} for $x = 0.15$ ceramics is largely due to its endurance under the higher electric field of 760 kV cm^{-1} . Yet, comparing Fig. 1a–1d, the maximum electric field applied are 700, 760, 715, and 695 for $x = 0.14, 0.15, 0.16,$ and $0.18,$ respectively. It is mentioned that the devices were measured up to their breakdown electric fields.
 - 3a. $x = 0.15$ seems unreasonably much higher than 0.14 and $0.16,$ resulting in the high W_{rec} of 14 J cm^{-3} for $x = 0.15$ against W_{rec} of $\leq 12 \text{ J cm}^{-3}$ for $x=0.14$ or 0.16 as extracted from Fig 1e–1g. Neither the evidence from the phase-transition nor the grain-size point of view supports much difference.
 - 3b. "Furthermore, the unipolar P-E hysteresis loops with $x \geq 0.14$ and $x \leq 0.12$ were measured up to their breakdown electric fields." This is confusing as nothing between 0.12 and 0.14 was measured. Please rephrase it and its context to help readers easier understand they are referring to Fig. 1 and S2, respectively.
4. In line 135, is it Fig. S2a–d instead of S2a–b?
5. It is not clearly mentioned how Fig. 3d is obtained in the manuscript.
 - 5a. Are the data points in Fig. 3d extracted from the peak value temperature of Fig. 3c?
 - 5b. If yes, given that the peaks in Fig. 3c are broad and flat for $x \geq 0.06$ (as mentioned in line 198), more details about the extraction method/process are needed. If no, please include more details about how Fig. 3d is obtained.
 - 5c. The logic from Fig. 3c to Fig. 3d (i.e. line 198–208) is unclear. For example, beside a and b above, how is "Moreover, the superior P_{max} in the current systems is closely related to the large dielectric constant" concluded?
6. In line 174–184, the authors use the coexistence of the O and T phases (pseudo-cubic phase) to explain the low P_r . Yet, comparing Fig. S1b and Fig. 3d, $x = 0.12$ is already in the O–T coexistence phase under room temperature while its P_r is still higher than $x = 0.03$ and 0.06 (which are in O phase according to Fig. 3d).
7. In line 225–233, the authors use average grain size (AGS) and porosity to explain the high E_b . Yet, there is some inconsistency between the figures. In Fig. S6, $x = 0.09, 0.12, 0.14, 0.15,$ and 0.16 all show similar AGS of $140\text{--}170 \text{ nm}$ (in contrast to 230 nm for $x = 0.06$ and 220 nm for $x = 0.18$). Comparing Fig. 1a–d and S2b–d, $x = 0.14, 0.15, 0.16,$ and 0.18 all show $E_b > 690 \text{ kV cm}^{-1}$ while $x = 0.03, 0.06, 0.09,$ and 0.12 all show $E_b < 380 \text{ kV cm}^{-1}$. This is inconsistent with Fig. S6 and the explanations.

Reviewer #3

(Remarks to the Author)

In this paper, the authors have adjusted the dopants in KNN dielectric ceramics to enhance the energy storage properties to approximately 14 J/cm^3 with an efficiency of about 89%. I believe these properties are suitable for publication in professional journals on energy storage, but not in Nature Communications due to a lack of new understanding or innovative strategies for enhancing energy storage properties. Additionally, the authors claim that the designed ceramics have numerous applications in electronic devices. Can the authors design an electronic protocol device as evidence to demonstrate the superior energy storage performance? There are some other detailed comments for the authors.

1. The introduction is very broad and not based on a careful analysis of the current developments in energy storage materials. For instance, there are many works on enhancing energy storage properties in dielectrics based on coexistent polar states. What distinguishes this research from other works in terms of fundamental innovation?
2. The authors claim that poor energy storage properties are linked to unsatisfactory microstructure. Could the authors provide clear guidance for achieving a satisfactory microstructure for superior storage properties? This is very important for the understanding of readers and the academic community. Please avoid simply mentioning the coexistent state, which has been referenced many times.
3. Why do the authors believe that a W_{rec} below 5.7 J/cm^3 will hinder integration into electronic devices? I think the important point is the actual energy storage output in electronic devices, especially under real working voltages like 220 V or lower.
4. What is the distinct role of the second compound and point defect in energy storage behavior?
5. Some papers reveal that rare-earth elements can improve the d_{33} , requiring large P_r after removing the field. Why does the P_r in Nd-doped KNN here behave differently?
6. Why did the authors initially measure the energy storage properties under a low field of 200 kV/cm ?
7. Why do the polars with high dynamics exhibit enhanced relaxor behavior? Why is higher relaxor behavior accompanied by low $t_{0.9}$?
8. How do the authors justify the assumption: "The origin is inferred to be the substitution of Nd^{3+} at the A-site, leading to the generation of vacancies and expansion of the oxygen octahedron"?

9. Why does the C phase in Fig. 5b display local polarization? I am concerned that the evidence of C/T/O coexistence is insufficient.

10. Why doesn't the phase field simulation show the coexistence of C/T/O states? I believe it is difficult to induce the C phase in this composition.

11. Why is the discussion part the conclusion?

Version 1:

Reviewer comments:

Reviewer #1

(Remarks to the Author)

The authors have addressed all the comments thoroughly and made meaningful revisions that enhance the manuscript's clarity and scientific rigor. I am a big fan of Table R1 to distinguish the contributions of each mechanism. Hence, I recommend accepting the revised manuscript after ensuring all changes are correctly integrated into the main text and supplementary materials.

Reviewer #3

(Remarks to the Author)

We appreciate the authors' careful responses and revisions to the manuscript. However, there are still two issues that need to be addressed before the manuscript can be considered for publication:

1. The authors have listed several papers reporting high energy storage densities, but these values are lower than the 14 J/cm³ reported in the current work. However, recent literature appears to include higher energy storage densities, such as 15.9 J/cm³ (Nature Communications, 2024, 15:7338) and 15.48 J/cm³ (Nature Communications, 2024, 15:6754). These references should be included in the manuscript, particularly in the introduction section and in Figure 2 (a), (b), and (h).

2. The authors describe three key factors contributing to the enhanced energy storage performance: (1) dense microstructure, (2) relaxor behavior, and (3) minimal impurities. However, the manuscript primarily discusses small grains and certain nanodomains, with supporting evidence for grain size provided in the supplementary materials. It remains unclear how the small grain size relates to the proposed heterogeneous strategy. This connection should be clarified in the manuscript.

Version 2:

Reviewer comments:

Reviewer #3

(Remarks to the Author)

The authors have addressed all my concerns and I now recommend the manuscript for acceptance.

We really appreciate the time and effort that the reviewers have invested in
evaluating our manuscript entitled “Excellent energy storage properties in lead-free
ferroelectric ceramics via heterogeneous structure design” (NCOMMS-24-49092). We
have carefully read the reviewers’ feedback, all the comments and suggestions are
important to improve the quality of our manuscript. We have revised our manuscript
accordingly, and the point-by-point response to the comments are enclosed in this
response letter. In order to make the changes easily readable, we highlighted our
revisions in the revised manuscript.

**Reviewer #1**

**General comments:**

This work presents a heterojunction design strategy aimed at further improving
energy storage performance, particularly at relatively low electric fields. Overall, the
research is systematic and of high quality, and I would not be surprised to see it
published in other high-impact journals. However, to meet the standards for Nature
Communications, the authors need to address the following points, particularly the
first one, before acceptance, as part of a major revision:

We sincerely appreciate your valuable feedback and suggestions on our
manuscript. According to your suggestions, the manuscript has been revised as
follows, which I have replied to point by point below. (All changes have been
presented with blue font in the revised manuscript)

1. Your question: SNZ dopant identification and heterojunction design: how is
Sr_{0.7}Nd_{0.2}ZrO₃ (SNZ) identified as a dopant in the KNN lattice? Is the improvement in

this work solely attributed to the heterojunction design? For instance, adding
external dopants to form a heterostructure can also involve other mechanisms such
as high-entropy, domain, defect, or grain size engineering. Please clarify how these
mechanisms are distinguished and discuss how the current design fits within or
diverges from them. It is critical to highlight the key differences, novelty, and
uniqueness of the approach to meet the acceptance standards of Nature
Communications.

**Response:** Thank you for your questions. We agree that further discussion is needed
on how to identify $\text{Sr}_{0.7}\text{Nd}_{0.2}\text{ZrO}_3$ is identified as a dopant in the KNN lattice is
necessary. The relevant discussions are as follows:

First, relaxor ferroelectric (RFE) materials have been recognized as great
potential candidates for energy storage optimization due to their nanodomains with
low energy barriers to polarization switching, rather than the strongly intercoupled
micrometer-sized domains found in ferroelectrics (FEs). The generally accepted
approach to realizing RFEs is to fabricate solid solutions combining FEs with
paraelectric end members, by which the long-range ferroelectric order is broken into
nanodomains. Some representative examples are $\text{BaTiO}_3\text{-BaZrO}_3$, $\text{BiFeO}_3\text{-SrTiO}_3$ and
$\text{Bi}_{0.5}\text{Na}_{0.5}\text{TiO}_3\text{-SrTiO}_3$ [*Adv. Mater.* **2017**, 29, 1604427; *Small* **2022**, 18, 2106515;
*Energy Environ. Sci.* **2020**, 13, 2938-2948; *Nat. Commun.* **2018**, 9, 1813; *ACS Appl.*
*Mater. Interfaces* **2023**, 15, 6990-7001]. The method that is most commonly
employed involves the introduction of rare earth ions to optimize the microstructure
[*Ceram. Int.* **2020**, 46, 21719-21727; *J. Eur. Ceram. Soc.* **2021**, 41, 352-359; *J. Eur.*

*Ceram. Soc.* **2017**, 37, 413-418; *J. Power Sources* **2024**, 624, 235548; *Ceram. Int.* **2022**,
48, 17359-17368]. Specifically, Nd ions were adopted by Wang et al. to improve the
energy storage performance of materials [*Energy Environ. Sci.* **2019**, 12, 582-588].
The incorporation of suitable Nd ions increased the insulation of grain boundaries
and reduced carrier mobility because the formed defect dipoles suppress the
movement of oxygen vacancies and then reduce the ion conductivity, leading to a
substantially enhanced E_b . Furthermore, compared with Ti^{4+} ions, Zr^{4+} ions are more
stable under an external electric field or a high temperature, which is in favor of a
low leakage current and hence a high E_b [*J. Mater. Chem. A* **2021**, 9, 18026-18085; *J.*
*Alloys Compd.* **2015**, 640, 416-420; *Nano Energy* **2018**, 52, 203-210; *J. Mater. Chem. C*
**2017**, 5, 9552-9558; *J. mater. Chem. C* **2018**, 6, 8528-8537]. Therefore, we developed
a binary system in which paraelectric end member, $Sr_{0.7}Nd_{0.2}ZrO_3$, is incorporated
into $K_{0.5}Na_{0.5}NbO_3$ matrix to enhance energy storage performance.

Second, the bottleneck in achieving energy storage performance by combining
FEs with paraelectric end members is the substantially sacrificed polarization.
Therefore, different approaches are desired to develop next-generation RFEs for
high-performance energy storage applications. We propose to achieve RFEs with
simultaneously high polarization and low loss by judiciously introducing polymorphic
nanodomains, e.g., orthorhombic (O) and tetragonal (T) nanodomains that coexist in
a cubic paraelectric matrix with competitive free energies. According to the phase
boundary regulation of KNN-based ceramics by previous studies [*Chem. Soc. Rev.*
**2020**, 49, 671-707; *Chem. Rev.* **2015**, 115, 2559-2595; *J. Appl. Phys.* **2007**, 102,

034102; *J. Am. Ceram. Soc.* **2010**, 93, 2783-2787; *J. Alloy. Compd.* **2016**, 672,
277-281], the ions in $\text{Sr}_{0.7}\text{Nd}_{0.2}\text{ZrO}_3$ are also considered as additives used to tailor $T_{\text{O-T}}$
and $T_{\text{T-C}}$ to form O-T-C multiphase nanoclusters coexisting at the local scale. In
addition, considering that Sr, Nd, and Zr ions can effectively suppress grain growth,
and enhance dielectric properties [*Chem. Eng. J.* **2020**, 390, 124566; *Energy Environ.*
*Sci.* **2019**, 12, 582-588], the incorporation of the secondary compound $\text{Sr}_{0.7}\text{Nd}_{0.2}\text{ZrO}_3$
into $\text{K}_{0.5}\text{Na}_{0.5}\text{NbO}_3$ ceramics is expected to yield high E_b , low dielectric loss, and
satisfactory polarization characteristics. This combination is conducive to achieving
excellent energy storage capabilities.

Third, we focused on dopants with ionic radii comparable to those of K^+ ,
Na^+ , and Nb^{5+} , as this facilitates effective substitution within the lattice. The ionic
radii of Sr^{2+} (1.44 Å, CN = 12), Nd^{3+} (1.27 Å, CN = 12), and Zr^{4+} (0.72 Å, CN = 6) are
expected to be compatible with the KNN lattice structure. Specifically, K^+ (1.64 Å, CN
= 12) and Na^+ (1.39 Å, CN = 12) can be effectively substituted by Sr^{2+} and Nd^{3+} , while
Zr^{4+} is able to replace Nb^{5+} (0.64 Å, CN = 6), indicating that SNZ can enter the main
lattice of KNN and form a perfect solid solution.

While the exceptional energy storage performance can be primarily attributed
to the heterogeneous structure, we acknowledge that other mechanisms also play a
significant role. For example, domain engineering optimizes switching dynamics (Figs.
4a-h), defect engineering modifies vacancies (Fig. 4j), and grain size engineering
refines grain size (Fig. S6, Supporting Information). We have included a comparative
table (**Table R1**) that distinguishes the contributions of each mechanism [*Science* 385,

**2024**, 204-209; *Science* **2019**, 365, 578-582; *Adv. Mater.* **2022**, 34, 2205787; *Nat. Rev.*
*Mater.* **2023**, 8, 355-356; *Nano Energy* **2023**, 112, 108458; *Nat. Energy* **2023**,
8, 956-964; *Adv. Energy Mater.* **2020**, 10, 1903338; *Energy Storage Mater.* **2020**, 30,
392-400; *J. Mater. Chem. C* **2024**, 12, 13343-13352; *Chem. Eng. J.* **2020**, 390, 124566],

while in the current research, the formation of heterogeneous structure is the
dominated mechanism responsible for the observed energy storage properties.

High-entropy material refers to a system containing at least five cations, the
atomic fraction is between 5-35 at%, and the configuration entropy is above 1.5R (R
is the ideal gas constant). The specific value of configuration entropy can be
calculated using the following formula:

$$\Delta S_{config} = -R \left[\left(\sum_{a=1}^n x_a \ln x_a \right)_{A-site} + \left(\sum_{b=1}^n x_b \ln x_b \right)_{B-site} + \right. \\ \left. 3 \left(\sum_{c=1}^n x_c \ln x_c \right)_{O-site} \right] \quad (1)$$

where x_a , x_b , and x_c are the mole fraction of the ions present in the A-site, B-site and
O-site, respectively [*J. Eur. Ceram. Soc.* **2018**, 38, 2318-2327; *Adv. Mater.* **2024**, 36,
2409059]. According to Eq. (1), the configuration entropy of KNN and
0.85KNN-0.15SNZ ceramics are 0.69R and 1.49R, respectively, showing that the KNN
and 0.85KNN-0.15SNZ ceramics do not belong to high-entropy materials.

1 **Table R1:** A comparative analysis of mechanisms in heterogeneous structures,
 2 high-entropy effects, domain engineering, defect engineering, and grain size
 3 engineering.

4

Mechanism	Description	Key Contributions to Energy Storage	Unique Features
Heterogeneous Structure (Main effect)	Coexistence of rhombohedral and/or orthorhombic, as well as tetragonal polar nanoregions within a cubic matrix	Enhance polarization while minimizing hysteresis, thereby elevating energy density and enhancing efficiency	Introduce polymorphic nanodomains
High-Entropy Effects (Negligible effect)	Incorporation of multiple principal elements in approximately equal proportions $\Delta S_{config} \geq 1.61R$	Stabilize multiple phases, enhance thermal and chemical stability, and broaden polarization, thereby enhancing energy storage performance	Composition of multiple elements and unique microstructure
Domain Engineering (Secondary Effect)	Manipulation of domain structures in ferroelectric materials	Optimize domain wall motion and switching dynamics, improving energy density and efficiency	Control domain sizes, orientation, and switching behavior
Defect Engineering (Secondary Effect)	Introduce and regulate defects in materials	Modify electronic and ionic conductivity	Act as charge traps or enhance carrier mobility
		Refine the grain size and domain size due to vacancy-related defect pinning, thereby enhancing overall energy storage performance	Regulate the oxygen vacancies
Grain Size Engineering (Secondary Effect)	Regulation of grain size and distribution in polycrystalline materials	Improve the breakdown strength, thereby increasing the energy storage density	Refine the grain size to submicron level

According to the feedback, we have modified the "Introduction" in our revised
manuscript as follows:

We developed a binary system incorporating the paraelectric end member
$\text{Sr}_{0.7}\text{Nd}_{0.2}\text{ZrO}_3$ into a KNN matrix to realize RFEs, which are promising candidates for
energy storage optimization due to their weakly intercoupled nanodomains.¹⁴ ...
Meanwhile, the rare earth ion of Nd^{3+} contributes to enhancing energy storage
performance by reducing grain size and optimizing the microstructure of dielectric
ceramics^{35,36}. ...This finding proves that a heterogeneous structure can be constructed
through a multi-scale process using simple chemical compositions and conventional
solid-phase reactions. These insights are invaluable for the development of
high-performance energy storage materials.

2. Your question: Breakdown voltage or field: What is the breakdown voltage or field
of this material? Please provide specific values, as this is crucial for understanding
the material's practical application limits and its performance under high electric
fields.

**Response:** Thanks for your question. Dielectric breakdown strength (DBS, E_b) also
known as breakdown electric field or dielectric strength is the maximum electric field
the dielectric material can withstand under the action of an electric field without
breakdown (that is, changing from an insulating state to a conductive state) [*Adv.*
*Funct. Mater.* **2023**, 33, 2301027; *J. Mater. Chem. A* **2023**, 11, 2641]. It is a crucial
parameter for dielectric materials, influencing their reliability and safety under high
electric fields. We have achieved the highest breakdown field of $\sim 760 \text{ kV cm}^{-1}$ in the

composition of $x = 0.15$.

According to the suggestion, we have added the specific values in the
introduction in our revised manuscript as follows:

As expected, we achieved a remarkable W_{rec} of $\sim 7 \text{ J cm}^{-3}$ with an outstanding η
of $\sim 92\%$ at a moderate electric field of 500 kV cm^{-1} and a giant W_{rec} of $\sim 14 \text{ J cm}^{-3}$
with a high η of $\sim 89\%$ at an electric field of 760 kV cm^{-1} , approaching E_b .

3. Your question: Energy efficiency vs. electric field: Please include a detailed
discussion on the relationship between energy efficiency and electric field. How does
the electric field affect energy efficiency, and what factors contribute to the observed
changes? A graph or a more explicit relationship between these parameters would
be beneficial to the readers.

**Response:** Thanks for your question. According to this suggestion, we have added
the relevant discussions as follows:

The energy efficiency increases initially and then decreases as a function of
electric field (Fig. 1e-h). This is mainly because ceramics exhibit low dielectric losses
and energy dissipation under low electric fields, and the movement of charge carriers
remains largely unaffected, maintaining low conductivity and reducing leakage
current. An increased electric field enhances polarization, improving energy storage
capacity and efficiency. However, under high electric fields, the loss and leakage
current increase because of the increased dielectric heating and nonlinear effect of
the dielectrics, meanwhile the field induce defects, thus increasing leakage current
[*Sens. Actuators, A* **2020**, 303, 111724; *Nat. Commun.* **2024**, 15, 702; *Nat. Mater.*

**2022**, 21, 1074-1080; *ACS Nano* **2022**, 16, 19959-19979]. And we have modulated
the “Results” in our revised manuscript as follows:

To further explore the properties of the ceramics, the energy storage
parameters (W_{rec} , W_{total} , and η) are depicted in Figs. 1e-h and S2e-h (Supporting
Information). Owing to the increasing polarization of ceramics, η generally improves
under low electric fields. However, under high electric fields, the dielectric loss and
leakage current increase because of the increased dielectric heating and nonlinear
effect of the dielectrics, meanwhile the field induces defects, thus increasing leakage
current, negatively affecting η (Fig.S3, Supporting Information).

We believe a graph illustrating these parameters would benefit readers.
Therefore, we have included a schematic (**Fig. R1**) in the Supplementary Materials
(Fig.S3, Supporting Information) as follows:

**Fig. R1** Schematic of the relationship between η and E .

4. Your question: SNZ doping ratio of up to 0.18: How was the optimal doping ratio
range of ~ 0.18 for SNZ in the KNN lattice identified? Please provide more details on
the experimental or theoretical justification behind this choice, along with a
discussion of the effects of varying the doping ratio on the material's energy storage
performance and efficiency.

**Response:** Thank you for your question and suggestion. We agree that it is necessary
to further enrich the discussion. And the relevant discussions are as follows:

First, the crystalline structures of (1-x)KNN-xABO₃ ceramics (where ABO₃ serves
as a secondary compound), as analyzed through X-ray diffraction (XRD), generally
indicate that a secondary phase tends to emerge at x=0.20, which will degrade the
energy storage performance. This observation suggests that due to variations in ionic
radius and valence, the solubility of ABO₃ within KNN ceramics is limited to less than
0.2 [*J. Mater. Chem. A* **2017**, 5, 554; *J. Mater. Chem. C* **2020**, 8, 8777; *Adv. Funct.*
*Mater.* **2022**, 32, 2111776].

Second, we implemented a heterogeneous structure strategy that pushed the
boundary of W_{rec} in ceramics with compositions of $x = 0.14$ (11.7 J cm⁻³), 0.15 (14.0 J
12 cm⁻³), 0.16 (11.4 J cm⁻³), and $x = 0.18$ (10.2 J cm⁻³). When x exceeds 0.15, both E_b and
13 P_{max} decrease as SNZ concentration increases, indicating that ceramics with $x = 0.15$
exhibit the optimized properties. Furthermore, the observed increase in average
grain size (AGS) and decrease in dielectric constant of $x = 0.20$ ceramics (as illustrated
in **Fig. R2**) further suggest a reduction in E_b and P_{max} , which consequently leads to a
decrease in W_{rec} . Therefore, it can be concluded that the best properties have been
achieved at $x = 0.15$, considering the W_{rec} beyond 10 J cm⁻³, we concluded the optimal
doping ratio range of SNZ within KNN for composition range is 0.14- 0.18. And we
have added the theoretical justification behind this choice in our revised manuscript
as follows:

**Additionally, for $x \geq 0.15$, both E_b and P_{max} decrease with increasing SNZ**

content, indicating that ceramics with $x = 0.15$ exhibit the best properties.
 Considering the W_{rec} beyond 10 J cm^{-3} , we conclude the optimal doping ratio range of
 SNZ within KNN for composition range is 0.14 - 0.18.

**Fig. R2** a) FE-SEM image of $x = 0.20$ ceramics. b) Temperature-dependent dielectric
 constant of $x = 0.20$ ceramics at various frequencies.

We have added the effects of varying the doping ratio on the material's energy
 storage performance and efficiency as follows:

A $W_{rec} < 5 \text{ J cm}^{-3}$ can be observed in a composition range of $x < 0.14$,
 accompanied by an $\eta < 80\%$. In contrast, ceramics with $x \geq 0.14$ consistently maintain
 superior energy storage performance with W_{rec} beyond 10 J cm^{-3} and simultaneously
 high efficiency about 90% under electric fields approaching their E_b values. Of
 particular significance is that a superior energy storage density of 14 J cm^{-3} with a
 high η of 89% have been achieved in composition of $x = 0.15$ ceramic under an
 electric field of 760 kV cm^{-1} .

5. Your question: Efficiency drop at higher electric fields (92% to 89%): Why does the
 energy efficiency drop from 92% to 89% at higher electric fields? Please discuss the
 potential causes, such as increased dielectric losses or structural limitations at higher

fields. Additionally, suggest possible mitigation strategies, such as optimizing grain
size or phase distribution, to maintain high efficiency even at elevated fields.

**Response:** Thank you for your question and suggestion. The reduced energy
efficiency at higher electric fields may be due to the following factors: 1) Dielectric
loss: Under high electric fields, the dielectric loss of materials can increase, resulting
in increased energy dissipation as heat, thereby reducing energy efficiency. 2)
Leakage current: Due to the non-linear effect of dielectrics under high field and the
field induces defects, thus increasing leakage current. 3) Changes in domain structure:
High electric fields can alter the domain structure, leading to domain wall movement
and merging of domains. This affects materials' polarization response and energy
efficiency.

We agree that we can maintain high efficiency at high electric field by
optimizing grain size or phase distribution, for the motivation of this research, we
keep experimental conditions the same for all studied composition to understand the
impact of SNZ component within KNN system for energy storage applications.
Following the suggestion, we have added discussion for the potential mitigation
strategies to increase the efficiency of the materials under high electric field, as
follows:

Even though the efficiency slightly decreases near the breakdown strength, it
can be further enhanced by optimization of grain size and phase distribution.

6. Your question: Phase composition (O-T-C) and absence of secondary phases: Is
there any evidence of rhombohedral phases, or is the material strictly limited to

orthorhombic, tetragonal, and cubic (O-T-C) phases? Also, why are no secondary
phases observed? This point could be clarified by further discussing the synthesis
process and the stability of the material under varying conditions.

**Response:** Thank you for your question and suggestion. In KNN-based energy storage
ceramics, the phase composition is predominantly characterized by orthorhombic (O),
tetragonal (T), and cubic (C) phases [*Chem. Rev.* **2015**, 115, 2559–2595; *Adv. Funct.*
*Mater.* **2021**, 2111776; *Energy Storage Mater.* **2022**, 45, 861-868]. The KNN phase
diagram indicates that the stability of the rhombohedral phase is limited compared
to the O-T-C phases. The rhombohedral phase can appear under certain composition
or temperature conditions, but it is typically not the dominant phase in most KNN
formulations for energy storage [*IEEE* **2006**, 201-207; *J. Am. Ceram. Soc.* **2013**, 96,
3677-3696]. In addition, the solid-state reaction method can significantly affect
phase formation. For instance, high-temperature sintering often stabilizes the
tetragonal and orthorhombic phases, while lower temperatures might allow for the
formation of rhombohedral structures. In this work, the raw specimens were placed
in covered alumina crucibles and sintered in the air in the temperature range of 1200
to 1300 °C for 5 h to fabricate dense ceramics and stabilize tetragonal and
orthorhombic phases. Additionally, Rietveld refinement XRD results, local
polarization configuration analysis, temperature-dependent dielectric constant
measurements, and phase-field simulations confirm that the material is strictly
limited to orthorhombic, tetragonal, and cubic (O-T-C) phases.

The absence of secondary phases can be attributed to the following reasons: 1)

Composition design: The effective design of components is crucial for regulating the
formation of secondary phases. The crystalline structures of (1-x)KNN-xSNZ ceramics
were detected by XRD, as depicted in Fig. S4 (Supporting Information). The results
indicate that all samples ($x \leq 0.18$) exhibit a single perovskite phase, confirming that
SNZ enters the main lattice of KNN and forms a perfect solid solution. Second
compound has a delimited solid solubility in KNN ceramics due to the difference in
ionic radius and valence state, Therefore, irrational component design (e.g., $x > 0.18$)
can result in secondary phase formation. 2) Synthesis process: a) Precise control of
synthesis parameters (temperature, rate, and time) ensures the formation of desired
phases while minimizing secondary phase formation. b) The burying sintering
process involved covering raw specimens with calcined powders in sealed alumina
crucibles to prevent alkali metal volatility and achieve a single-phase structure. c)
Secondary ball milling was performed to reduce particle size and increase specific
surface area, optimizing sintering and minimizing second phase formation. We have
included related discussions in our revised manuscript as follows:

It reveals the coexistence of the orthorhombic (O, $Amm2$) and tetragonal (T,
$P4mm$) phases, with weight fractions of 26.14% and 73.86%, respectively. The
absence of rhombohedral phases, which can appear under certain composition or
temperature conditions, is due to high-temperature sintering that stabilizes the T and
O phases. Notably, no secondary phase is observed, indicating the complete
incorporation of the SNZ into the KNN lattice (Fig. S6a, Supporting Information),
which can be attributed to optimized synthesis processes and rational composition

design.

7. Your question: Material system reliability and stress testing: What is the reliability
of this material system over time? To assess its practical use in real-world
applications, please consider adding stress tests to check whether energy storage
properties degrade under mechanical, thermal, or electrical stress. This will provide
insights into the material's long-term performance and robustness.

**Response:** Thank you for your question and suggestion. Due to the limitation of the
laboratory, it is difficult to obtain the cycling reliability as a function of preload stress.

In addition, energy storage application, it is usually used in MLCC, which will not
operate under preload or prestress. Excellent cycling/frequency stability of the
energy storage performance would give the capacitors an enormous application
range [*Adv. Funct. Mater.* **2022**, 32, 2110478]. The *P-E* loops, which show negligible
changes over 10^5 cycles and frequencies ranging from 5 to 400 Hz at 200 kV cm^{-1} ,
shown in **Fig. R3a** and **Fig. R3c**, confirm superior cycling reliability ($W_{\text{rec}} \sim 1.77 \pm 0.02 \text{ J}$
15 cm^{-3} , $\eta \sim 90.0 \pm 2.0\%$) and frequency stability ($W_{\text{rec}} \sim 1.67 \pm 0.07 \text{ J cm}^{-3}$, $\eta \sim 89.5 \pm 2.5\%$)
for $x = 0.15$ ceramics, as illustrated in **Fig. R3b** and **Fig. R3d**, respectively. The $x = 0.15$
ceramics exhibit not only ultrahigh energy storage performance but also excellent
frequency stability and fatigue resistance, making them highly suitable for practical
application in cutting-edge capacitors.

**Fig. R3** Fatigue resistance of (a) P - E loops, and (b) W_{rec} and η values with respect to
 cycling numbers at an electric field of $200\text{ kV}\cdot\text{cm}^{-1}$ of $x = 0.15$ ceramic. (c) P - E loops,
 and (d) W_{rec} and η values with respect to frequency at an electric field of $200\text{ kV}\cdot\text{cm}^{-1}$
 of $x = 0.15$ ceramic.

We have included Fig. S5 (**Fig. R3**) along with relevant discussions in the
 Supporting Information section. The following discussion has been added to the
 revised manuscript:

In addition, the superior stability (Fig. S5, Supporting Information) of this
 material enhance its practical application in demanding energy storage conditions.

8. Your question: SNZ ratio and its effect on P_{max} and P_r in Figure S1(b): In Figure S1(b),
 why does the SNZ ratio initially cause the maximum polarization (P_{max}) and remnant
 polarization (P_r) to increase but decrease after a certain point? Please explain the
 underlying mechanism, especially around the observed maximum, and discuss the

correlation between the SNZ ratio and energy efficiency.

**Response:** Thank you for your question and suggestion. In Figure S1(b), it can be
seen that with an increase of the SNZ concentration, both P_{\max} and P_r increases first
and then decreases, indicating that the both P_{\max} and P_r are the highest when the
dissolved content of SNZ is 0.09 mol. The underlying mechanism can be explained as
follows: 1) Low doping level ($x < 0.09$): a) The presence of A-site vacancies induced by
the introduction Sr^{2+} and Nd^{3+} (aliovalent substitution) is conducive to increasing the
ϵ_r (Fig. 3d) and thus generating a high P_{\max} [*J. Mater. Chem. C* **2019**, 7, 4999-5008; *J.*
*Mater. Chem. A* **2018**, 6, 17896-17904; *J. Mater. Chem. A* **2021**, 9, 18026-18085]. b)
The ionic radii of Sr^{2+} (1.44 Å, CN = 12), Nd^{3+} (1.27 Å, CN = 12) are smaller than those
of K^+ (1.64 Å, CN = 12) and Na^+ (1.39 Å, CN = 12), resulting in a more facile
displacement of ions at the B-site and enhanced polarization. c) The ceramics with x
≤ 0.09 exhibit a single O phase, resulting in high polarization. 2) High doping level ($x >$
0.09): a) With increasing SNZ, the long range order is disrupted and induce relaxor
components, as shown in Fig. 3d, accounting for the reduced P_r . b) SNZ, as a
paraelectric end member has the low ϵ_r around room temperature together with the
lack of spontaneous polarization, corresponding well to the reduced P_{\max} at high
doped level [*Nano Energy* **2023**, 109, 108275]. Therefore, the ceramics of $x = 0.09$
exhibit maximum values of P_{\max} and P_r , which can be attributed to factors such as
ionic radius, A-site vacancies, and phase structure.

In this work. we developed a binary system in which paraelectric end member,
SNZ, is incorporated into KNN matrix. Therefore, while certain components may

display minor deviations due to variations in microstructure, material properties,
uniformity of electric field distribution, breakdown strength, and saturation
polarization, the overall energy efficiency tends to improve with an increase in SNZ [*J.*
*Mater. Chem. A* **2021**, 9, 18026-18085].

9. Your question: Scalability: Will larger areas increase leakage and affect energy
storage performance? Is the randomness in the heterogeneous structure sufficient to
cover leakage variability, or is there still an area dependency that impacts normalized
energy storage efficiency? Addressing this would clarify the material's suitability for
large-scale applications.

**Response:** Thank you for your question and suggestion.

First, in this study, all the samples were electroded with fired-on silver paste
with an area of about 0.785 mm² (1 mm in diameter), for the energy storage
properties evaluation. The electrode area is commonly used and consistent with
literature studies published in peer-reviewed journals, which has been adopted to be
a standard for energy storage performance comparisons between various material
systems [*Adv. Mater.* **2022**, 34, 2204356; *J. Am. Chem. Soc.* **2023**, 145, 6194–6202;
*Adv. Mater.* **2024**, 36, 2313285; *Adv. Funct. Mater.* **2023**, 33, 2301027].

Second, E_b is primarily determined by factors such as relative density, grain size,
porosity, band gap, dielectric constant, dielectric loss, secondary phase, among
others [*J. Mater. Chem. A* **2022**, 10, 14316; *Small* **2023**, 19, 2303915; *Chem. Eng. J.*
**2022**, 446, 137389]. Factors including sample thickness, electrode
area/thickness/type, humidity, temperature, type of voltage, and test duration also

influence E_b , and thus affecting the energy storage performance. The probability of
leakage currents and defect density generally increases with increasing the testing
area, although the randomness in a heterogeneous structure can help distribute
defects and variations, potentially balancing out leakage currents, the breakdown
field will certainly degrade with increasing the area because of more porosity exist,
thus reduce the energy storage density.

**Reviewer #2**

**General comments:**

In this manuscript, the authors realized KNN-SNZ based, lead free dielectric
capacitors with outstanding energy storage performance. For explanations of the
enhanced performance, multi-phase coexistence is proposed and supported by both
experiment and simulation. The manuscript is well organized, and the evidence for
the explanation is to some degree comprehensive. On the other hand, there are
weak points in the manuscript, especially the inconsistency of the evidence across
the figures. To improve the overall quality, the following questions and comments
need to be addressed:

**Response:** Thanks so much for your constructive comments. We really appreciate
this and cherish this opportunity to broaden our knowledge and to improve
ourselves. According to your suggestions, the manuscript has been revised as follows,
which I have replied to point by point below. (All changes have been presented with
purple font in the revised manuscript)

1. Your questions: Please include the importance of being lead-free in this material
system. The manuscript only mentions the comparison in performance perspective
but doesn't convince readers why being lead-free is important.

**Response:** Thanks so much for your reminding and suggestion. According to your
suggestions, we have added the importance of the lead-free in "Introduction". The
related description in our manuscript as follows:

This deficiency is primarily attributed to their unsatisfactory microstructure,

rendering them unsuitable for advanced electronic applications⁵⁻⁷, but for
environmental and health considerations, lead-based candidates are strictly limited
for sustainable social development. Therefore, numerous efforts have been made to
improve the performance of lead-free ceramics for energy storage dielectric
capacitors⁸.

2. Your questions: What are the size and thickness of the devices? Does the
performance of the devices depend on the device size?

**Response :** Thanks for your reminding and question. And the former question has
been discussed as follows:

The samples for energy storage evaluation are 0.1 mm in thickness with fired-on
silver electrode 1 mm in diameter. To characterize their dielectric constant, the
samples were ground and polished to a thickness of 0.7 mm and electroded with
fired-on silver paste with the entire sample surface, 8.7 mm in diameter.

The energy storage performance depends on the device size, because the
energy storage density is proportional to the applied electric field square, which is
closely associated with the dielectric breakdown field. E_b is primarily determined by
factors such as relative density, grain size, core-shell structure, band gap, dielectric
constant, dielectric loss, defects, secondary phase, space charge, among others.
While factors including sample thickness, electrode area/thickness/type, humidity,
temperature, type of voltage, and test duration also play a significant role in affecting
E_b [*J. Mater. Chem. A* **2022**, 10, 14316; *Small* **2023**, 19, 2303915; *Chem. Eng. J.* **2022**,
446, 137389; *Chem. Eng. J.* **2023**, 467, 143395]. This information has been detailed in

the “Ceramics preparation” section.

3. Your questions: The record-high W_{rec} of 14 J cm^{-3} for $x = 0.15$ ceramics is largely
due to its endurance under the higher electric field of 760 kV cm^{-1} . Yet, comparing Fig.
1a-1d, the maximum electric field applied are 700, 760, 715, and 695 for $x = 0.14,$
0.15, 0.16, and 0.18, respectively. It is mentioned that the devices were measured up
to their breakdown electric fields. 3a. $x = 0.15$ seems unreasonably much higher than
0.14 and 0.16, resulting in the high W_{rec} of 14 J cm^{-3} for $x = 0.15$ against W_{rec} of $\leq 12 \text{ J}$
8 cm^{-3} for $x = 0.14$ or 0.16 as extracted from Fig 1e-1g. Neither the evidence from the
9 phase-transition nor the grain-size point of view supports much difference. 3b.
“Furthermore, the unipolar P - E hysteresis loops with $x \geq 0.14$ and $x \leq 0.12$ were
measured up to their breakdown electric fields.” This is confusing as nothing
between 0.12 and 0.14 was measured. Please rephrase it and its context to help
readers easier understand they are referring to Fig. 1 and S2, respectively.

**Response:** Thanks for your reminding and question. One of the highlights of this
study is the implementation of a more refined tuning of the composition, specifically
at $x = 0.14, 0.15,$ and 0.16. This precise control over the chemical composition of the
material aims to achieve optimal energy storage performance and offer deeper
insights into the interrelationships among composition design, structure, and
properties. The $x = 0.15$ bulk ceramics achieve exceptional energy storage capabilities.
The narrow tuning of the composition leads to similar phase structures and grain
sizes for 0.14, 0.15, and 0.16. However, the grain size in the $x = 0.15$ sample exhibits
greater uniformity (Fig S6), and the O to T phase ratio is optimized, as confirmed by

the polarization characteristics derived from the P - E loops under the same electric
 field of 700 kV cm^{-1} (**Fig. R4**). The sample with $x = 0.15$ exhibits higher ΔP and W_{rec} .

**Fig. R4** (derived from the Fig. 1a-c) The polarization and energy-storage properties of
 $(1-x)\text{KNN-xSNZ}$ ($x = 0.14, 0.15, 0.16$) ceramics at an electric field of 700 kV cm^{-1} .

Additionally, the impedance spectroscopy data for $(1-x)\text{KNN-xSNZ}$ ($x = 0.14, 0.15,$
 0.16) ceramics is tested to analyze the potential reasons for optimization of energy
 storage performances, as shown in **Fig. R5**. With an increase of the SNZ
 concentration, the total resistance increases first and then decreases, symbolizing
 that the total resistivity is the highest when $x = 0.15$ attributed to its high uniformity
 of the grain size, corresponding to the highest E_b . For $x = 0.14, 0.15,$ and $0.16,$ the
 maximum applied electric fields are $700, 760,$ and 715 kV cm^{-1} respectively, the
 variation between these three compositions are small ($\leq 60 \text{ kV cm}^{-1}$), due to their
 similar chemical composition and structural characteristics. It is noteworthy that the
 energy storage performance ($W_{\text{rec}} \geq 10 \text{ J cm}^{-3}$) of $0.14, 0.16,$ and even 0.18 ceramics
 is also decent.

**Fig. R5.** The comparison of impedance performance among $x = 0.14$, $x = 0.15$, and $x =$
 0.16 at $400\text{ }^{\circ}\text{C}$.

Again, we apologize for the unclear description in the original manuscript. After
 thorough review, the manuscript has been revised as follows:

This is attributed to their slender P - E shapes (Fig. S1a, Supporting Information)
 and significantly reduced P_r (Fig. S1b, Supporting Information). Furthermore, the
 unipolar P - E hysteresis loops of the ceramics were measured up to their respective
 breakdown electric fields. The results are illustrated in Figs. 1a-d ($x \geq 0.14$) and S2a-d
 (Supporting Information, $x < 0.14$). The slender unipolar P - E loops of KNN-xSNZ
 ceramics ($x = 0.14, 0.15, 0.16$, and 0.18 ; see Figs. 1a-d) under various electric fields
 showcase the canonical relaxor behaviors.

4. Your questions: In line 135, is it Fig. S2a-d instead of S2a-b?

**Response:** Thanks so much for your reminding and suggestion. We are sorry for our
 inaccurate description in the original manuscript. After careful checking, the
 manuscript has been revised as follows:

This can be attributed mainly to the high ΔP and the delayed polarization
 saturation (compare Fig. 1a-d to S2a-b, Supporting Information), demonstrating a

promising candidate for practical application in the advanced energy storage
capacitors.

5. Your questions: It is not clearly mentioned how Fig. 3d is obtained in the
manuscript. 5a. Are the data points in Fig. 3d extracted from the peak value
temperature of Fig. 3c? 5b. If yes, given that the peaks in Fig. 3c are broad and flat
for $x \geq 0.06$ (as mentioned in line 198), more details about the extraction
method/process are needed. If no, please include more details about how Fig. 3d is
obtained. 5c. The logic from Fig. 3c to Fig. 3d (i.e. line 198-208) is unclear. For
example, beside a and b above, how is “Moreover, the superior P_{\max} in the current
systems is closely related to the large dielectric constant” concluded?

**Response:** Thanks for your reminding and question. The data points (T_{T-C} and T_{O-T}) in
Fig. 3d are extracted from the dielectric anomaly peaks recorded at a frequency of
1 kHz of Fig. 3c. Although the ceramics with $x \geq 0.06$, as typical relaxor ferroelectrics,
exhibit broad and flat peaks as shown in Fig. 3c, the data points (T_{T-C} and T_{O-T}) can be
derived from the dielectric anomaly peaks corresponding to the recorded maximum
values. Further details of the extraction method/process are shown in the schematic
figure (**Fig. R6**). Besides, compared with other lead-free energy storage ceramics [*Adv.*
*Funct. Mater.* **2023**, 33, 2301027; *Adv. Mater.* **2024**, 2313285; *Nat. Commun.* **2020**,
11, 4824], the introduction of SNZ makes the dielectric constant of in the current
systems keep a high level, which is conducive to achieving high polarization under
the action of an electric field.

The logic from Fig. 3c to Fig. 3d (i.e. line 198-208) is indeed unclear. Following

the feedback, we have added the relevant discussions as follows:

The temperature-dependent dielectric constant at different frequencies of all
components (Fig. 3c) is consistent with the above analysis. Three main characteristics
can be derived from the dielectric properties: 1) The presence of an extremely broad
and flat dielectric peak with frequency dispersion behavior ($x \geq 0.06$, see Fig. 3c),
which can be further verified by the inset in Fig. 3c for $x = 0.15$ sample, represents a
canonical relaxor feature attributed to the presence of PNRs⁸. 2) The relatively high
room temperature dielectric constant (≥ 1000) observed in the current systems ($x \geq$
0.06) is conducive to achieving high P_{\max} . 3) It can be noted that all samples exhibit
two dielectric anomaly peaks (temperature denoted by " T_{O-T} " and " T_{T-C} "), which
represent the transitions from O to T phase and from T to C phase, respectively, upon
heating¹². The T_{O-T} and T_{T-C} data points of $(1-x)\text{KNN-xSNZ}$ ceramics at a frequency of
1 kHz are extracted to construct a phase diagram to identify the composition
dependence of the phase structures, as depicted in Fig. 3(d). It can be observed that
both T_{O-T} and T_{T-C} decrease with increasing x , leading to the formation of a coexisting
O-T phase in the composition of $x \geq 0.12$.

22 **Fig. R6** Temperature-dependent dielectric constant of the ceramics at 1 kHz.

6. Your questions: In line 174-184, the authors use the coexistence of the O and T
phases (pseudo-cubic phase) to explain the low P_r . Yet, comparing Fig. S1b and Fig.
3d, $x = 0.12$ is already in the O-T coexistence phase under room temperature while
its P_r is still higher than $x = 0.03$ and 0.06 (which are in O phase according to Fig. 3d).

**Response:** Thanks for your question. We believe that the coexistence of the O and T
phases (pseudo-cubic phase) is conducive to achieving low P_r , due to the fact that the
O/T local heterogeneities disrupt the long-range order and induce relaxor behavior in
the studied material system. However, it is important to note that the low P_r value
cannot be solely attributed to the phase structure. In Figure S1(b), it can be seen that
as the SNZ concentration increases, both P_{\max} and P_r first increase and then decrease,
indicating that both P_{\max} and P_r are highest under an electric field of 200 kV cm^{-1}
when the dissolved content of SNZ is 0.09 mol. The underlying mechanism can be
explained as follows: 1) Low doping level ($x < 0.09$): a) The presence of A-site
vacancies induced by the introduction Sr^{2+} and Nd^{3+} (aliovalent substitution) is
conducive to increasing the ϵ_r (Fig. 3d) and thus generating a high P_{\max} [*J. Mater.*
*Chem. C* **2019**, 7, 4999-5008; *J. Mater. Chem. A* **2018**, 6, 17896-17904; *J. Mater. Chem.*
*A* **2021**, 9, 18026-18085]. b) The ionic radii of Sr^{2+} (1.44 Å, CN = 12), Nd^{3+} (1.27 Å, CN
= 12) are smaller than those of K^+ (1.64 Å, CN = 12) and Na^+ (1.39 Å, CN = 12),
resulting in a more facile displacement of ions at the B-site and enhanced
polarization. c) The ceramics with $x \leq 0.09$ exhibit a single O phase, resulting in high
polarization. 2) High doping level ($x > 0.09$): a) With increasing SNZ, the long range
order is disrupted and induce relaxor components, as shown in Fig. 3d, accounting

for the reduced P_r . b) SNZ, as a paraelectric end member has the low ϵ_r around room
temperature together with the lack of spontaneous polarization, corresponding well
to the reduced P_{\max} at high doped level [*Nano Energy* **2023**, 109, 108275]. Compared
to $x = 0.09$ (single O phase), the P_r value for $x = 0.12$ (O-T coexistence phase) shows a
decrease.

7. Your questions: In line 225-233, the authors use average grain size (AGS) and
porosity to explain the high E_b . Yet, there is some inconsistency between the figures.
In Fig. S6, $x = 0.09, 0.12, 0.14, 0.15,$ and 0.16 all show similar AGS of 140-170 nm (in
contrast to 230 nm for $x = 0.06$ and 220 nm for $x = 0.18$). Comparing Fig. 1a-d and
S2b-d, $x = 0.14, 0.15, 0.16,$ and 0.18 all show $E_b > 690 \text{ kV cm}^{-1}$ while $x = 0.03, 0.06,$
$0.09,$ and 0.12 all show $E_b < 380 \text{ kV cm}^{-1}$. This is inconsistent with Fig. S6 and the
explanations.

**Response:** Thanks for your question. as discussed in our manuscript (in line 231),
refined AGS and reduced porosity have a favorable impact on E_b . However, it is
important to note that the high E_b value cannot be solely attributed to the grain size.
In addition, AGS and E_b exhibit some inconsistencies, primarily due to the significant
impact of polarization saturation on E_b . Many relaxor ferroelectrics, such as the
samples with $x < 0.14$ in the current system, although exhibit high ΔP , the slope of
the P - E loop rapidly decreases with increasing electric field, resulting in polarization
saturation at lower electric fields (Fig. S2), which limits the enhanced of E_b [*Nat.*
*Commun.* **2024**, 15, 5232; *J. Mater. Chem. A* **2024**, 12, 9124-913; *Science* **2022**, 375,
1418-1422].

**Reviewer #3**

**General comments:**

In this paper, the authors have adjusted the dopants in KNN dielectric ceramics
to enhance the energy storage properties to approximately 14 J/cm³ with an
efficiency of about 89%. I believe these properties are suitable for publication in
professional journals on energy storage, but not in Nature Communications due to a
lack of new understanding or innovative strategies for enhancing energy storage
properties. Additionally, the authors claim that the designed ceramics have
numerous applications in electronic devices. Can the authors design an electronic
protocol device as evidence to demonstrate the superior energy storage
performance? There are some other detailed comments for the authors.

**Response:** Thank you for your positive feedback on energy storage performance and
valuable suggestions for innovative strategies and electronic protocol devices. We
respectfully disagree with the assertion that our research lacks new insights or
innovative strategies for improving energy storage properties. In this regard, we
would like to emphasize the novel understanding and innovative approaches
presented in our work, along with the breakthroughs we've made, which we hope
you will find compelling. 1) Novel understanding: This study provides a novel
perspective on the relationship between microstructural characteristics (from grain
boundaries to domains, and then to O-T-C multiphase nanoclusters) and energy
storage properties, highlighting the importance of SNZ within the KNN system. 2)
Innovative approaches: This finding proves that a heterogeneous structure,

confirmed by aberration-corrected scanning transmission electron microscopy and
phase-field simulations, can be constructed through a multi-scale process using
simple chemical compositions and conventional solid-phase reactions. 3)
Breakthrough: We achieved a remarkable W_{rec} of $\sim 7 \text{ J cm}^{-3}$ with an outstanding η of \sim
92% at a moderate electric field of 500 kV cm^{-1} and a giant W_{rec} of $\sim 14 \text{ J cm}^{-3}$ with a
high η of $\sim 89\%$ at an electric field of 760 kV cm^{-1} , approaching E_b . This represents the
highest level of performance achieved in KNN-based bulk ceramics.

Regarding electronic protocol devices, we acknowledge the importance of
demonstrating practical applications. Unfortunately, due to time and resource
limitations, we are unable to develop and present a fully functional electronic
protocol device at this stage. Nevertheless, we have established collaborations with
other research groups, and the design of an electronic protocol device will be
included in our future work.

We sincerely appreciated for your valuable feedbacks and detailed comments
on our manuscript. According to your comments, the manuscript has been revised as
follows, which I have replied to point by point below. (All changes have been
presented with **green font** in the revised manuscript).

1. Your questions: The introduction is very broad and not based on a careful analysis
of the current developments in energy storage materials. For instance, there are
many works on enhancing energy storage properties in dielectrics based on
coexistent polar states. What distinguishes this research from other works in terms of
fundamental innovation?

**Response:** Thank you for your valuable feedback on the introduction. I appreciate
your point about needing a more focused analysis of recent developments in energy
storage materials. We have revised the “Introduction” and “Supporting Information”
to include a more thorough analysis of recent advancements in relaxor ferroelectric
ceramics, as detailed below:

In the “Introduction” section: A comprehensive analysis of the current
developments in the aforementioned energy storage materials can be found in the
Supporting Information. In the “Supporting Information” section: Relaxor
ferroelectric ceramics, in contrast to typical ferroelectrics, exhibit slimer P - E loops,
which facilitates the simultaneous enhancement of W_{rec} and η . Among these, BNT, BF,
BT, and KNN-based lead-free ceramics have emerged as research hotspots. Tables
S1-4 provide a comprehensive summary of the majority of BNT, BF, BT, and
KNN-based relaxor ferroelectric ceramics published in prestigious journals over the
past five years. Although various methods have demonstrated effectiveness in
improving energy storage properties, the currently achievable synergistic values of
W_{rec} and η remain unsatisfactory due to the inherent negative correlation between E_b
and ΔP , especially via traditional solid-state sintering methods. Furthermore, the
improved energy storage properties of bulk ceramics based on BNT, BF, BT, and KNN,
achieved through various optimization strategies, have been included in Tables S1-4
(Supporting Information). This amendment will provide context for our research
within the existing literature and underscore the specific areas we are addressing.

Indeed, several studies have focused on enhancing the energy storage

properties of dielectrics through the utilization of coexisting polar states. For instance,
Chen, L. et al. proposed a high-entropy strategy to design “local polymorphic
distortion” including R-O-T-C multiphase nanoclusters and random oxygen
octahedral tilt, resulting in enhanced energy storage properties ($W_{\text{rec}} \sim 10.06 \text{ J cm}^{-3}$, η
$\sim 90.8\%$) [*Nat. Commn.* **2022**, 13, 3089] were achieved in the chemical composition of
$[(\text{K}_{0.2}\text{Na}_{0.8})_{0.8}\text{Li}_{0.08}\text{Ba}_{0.02}\text{Bi}_{0.1}](\text{Nb}_{0.68}\text{Sc}_{0.02}\text{Hf}_{0.08}\text{Zr}_{0.1}\text{Ta}_{0.08}\text{Sb}_{0.04})\text{O}_3$. In the same year, they
introduced an effective strategy for constructing local diverse polarization in
superparaelectrics to design a new lead-free perovskite with a high $W_{\text{rec}} \sim 10.59 \text{ J cm}^{-3}$
and a large $\eta \sim 87.6\%$. The multiple polarization configurations of R-O-T-C were
developed using the chemical component of $1/3\text{BaTiO}_3$ - $1/3\text{Bi}_{0.5}\text{Na}_{0.5}\text{TiO}_3$ - $1/3\text{NaNbO}_3$
(BaTiO_3 : T phase; $\text{Bi}_{0.5}\text{Na}_{0.5}\text{TiO}_3$: R phase; and NaNbO_3 : O phase) [*Adv. Mater.* **2022**,
34, 2205787]. Additionally, Zeng, X. et al. achieved a large W_{rec} of 10.46 J cm^{-3} and a
high η of 86.25% in $0.94[0.6(\text{Bi}_{0.47}\text{Na}_{0.47}\text{Yb}_{0.03}\text{Tm}_{0.01})\text{TiO}_3$ - $0.4(\text{Ba}_{0.5}\text{Sr}_{0.5})\text{TiO}_3]$ -
$0.06\text{Sr}(\text{Zr}_{0.5}\text{Hf}_{0.5})\text{O}_3$ ceramics by designing dual-phase (R and T) coexistence and
induced oxygen octahedral tilt, combined with the tape-casting process [*Adv. Mater.*
**2024**, 36, 2409059]. In contrast, we have developed a heterogeneous structure that
is constructed through a multi-scale process from refining grain boundaries to
domains, and then to O-T-C multiphase nanoclusters in a relaxor with relatively
simple chemical composition of $(1-x)\text{K}_{0.5}\text{Na}_{0.5}\text{NbO}_3$ - $x\text{Sr}_{0.7}\text{Nd}_{0.2}\text{ZrO}_3$ ($(1-x)\text{KNN}$ - $x\text{SNZ}$)
using a conventional solid-phase reaction method. Notably, the 0.85KNN - 0.15SNZ
bulk ceramics achieve exceptional energy storage capabilities ($W_{\text{rec}} \sim 14 \text{ J cm}^{-3}$, η
$\sim 89\%$). The “Introduction” has been revised as follows:

However, the current situation indicates that the construction of such a
heterogeneous structure generally requires relatively complex chemical components
and/or processes. ...This finding proves that a heterogeneous structure can be
constructed through a multi-scale process using relatively simple chemical
compositions and conventional solid-phase reactions. Such insights are invaluable for
the development of high-performance energy storage materials.

2. Your questions: The authors claim that poor energy storage properties are linked
to unsatisfactory microstructure. Could the authors provide clear guidance for
achieving a satisfactory microstructure for superior storage properties? This is very
important for the understanding of readers and the academic community. Please
avoid simply mentioning the coexistent state, which has been referenced many
12 times.

**Response:** Thank you for your insightful comments concerning the relationship
between microstructure and energy storage properties. In our original manuscript,
we highlighted several key strategies that can enhance energy storage performance,
which we summarize as follows: 1) Dense microstructure: A dense microstructure
minimizes porosity and defects, which are crucial for achieving a high E_b value. 2)
Relaxor behavior: Relaxor ferroelectrics (RFEs) are proposed by introducing chemical
heterogeneity, which allows short-range ordered polar nanoregions (PNRs). The
formation of PNRs plays a pivotal role in reducing polarization switching barriers,
thus decreasing P_r . 3) Minimal impurities: Reducing impurities through careful
component design, high-purity raw materials, and controlled synthesis environments

can significantly enhance W_{rec} by minimizing defects in the crystal lattice.

Additionally, other microstructures also significantly affect energy storage
properties, such as nanodomain/domain [*Science* 385, **2024**, 204-209; *Science* **2019**,
365, 578-582; *Adv. Energy Mater.* **2020**, 10, 1903338], defect [*Energy Storage Mater.*
2020, 30, 392-400; *J. Mater. Chem. C* **2024**, 12, 13343-13352], grain size [*Nat. Rev.*
*Mater.* **2023**, 8, 355-356; *Chem. Eng. J.* **2020**, 390, 124566], and heterogeneous
structure [*Energy Environ. Sci.* **2023**, 16, 4511; *Adv. Mater.* **2022**, 34, 2205787; *Energy*
*Storage Mater.* **2022**, 45, 541-567]. To clarify these aspects, we have included a
comparative table (**Table R2**) that distinguishes the contributions of each
microstructure to energy storage performance. While the exceptional energy storage
performance can be primarily attributed to the heterogeneous structure, other
microstructural features also play a significant role. For instance, a dense structure,
refined grain size, small-sized nanodomains, vacancy-related defect pinning, and
stabilization of multiple phases without impurities all contribute to the overall
performance.

In this work, we propose an innovative design strategy for lead-free relaxors,
characterized by a heterogeneous structure that is constructed through a multi-scale
process from refining grain boundaries to domains, and then to O-T-C multiphase
nanoclusters, involving a combinatorial microstructure optimization process such as
refining grain/domain sizes, introducing coexisting nanoregions, inducing a
high-symmetry phase structure while minimizing impurities, and achieving a dense
microstructure with reduced porosity. The manuscript has been revised as follows:

High W_{rec} can be achieved by designing ceramics with fine grains, a dense
 microstructure, minimal impurities, high polarization under external electric fields, a
 highly dynamic structure, and an isotropic lattice structure²⁵⁻²⁸.

**Table R2:** A comparative analysis of microstructures in heterogeneous structures,
 domain structures, defect structures, and grain size.

Microstructure	Description	Key Contributions to Energy Storage	Unique Features
Heterogeneous Structure	Coexistence of rhombohedral and/or orthorhombic, as well as tetragonal polar nanoregions within a cubic matrix	Enhance polarization while minimizing hysteresis, thereby elevating energy density and enhancing efficiency	Introduce polymorphic nanodomains
Domain Structure	Manipulation of domain structures in ferroelectric materials	Optimize domain wall motion and switching dynamics, improving energy density and efficiency	Control domain sizes, orientation, and switching behavior
Defect Structure	Introduce and regulate defects in materials	Modify electronic and ionic conductivity	Act as charge traps or enhance carrier mobility
		Refine the grain size and domain size due to vacancy-related defect pinning, thereby enhancing overall energy storage performance	Regulate the oxygen vacancies
Grain Size	Regulation of grain size and distribution in polycrystalline materials	Improve the breakdown strength, thereby increasing the energy storage density	Refine the grain size to submicron level

6 3. Your questions: Why do the authors believe that a W_{rec} below 5.7 J/cm³ will hinder

integration into electronic devices? I think the important point is the actual energy
storage output in electronic devices, especially under real working voltages like 220 V
or lower.

**Response:** Thanks for your reminding and question. We greatly appreciate your
attention to the details of our manuscript. We apologize for the inaccurate
description in the original manuscript. Capacitors with any energy storage density
can be integrated into electronic devices. In our original manuscript, we aim to
convey that the materials published in peer-reviewed journals show exceptional
energy storage performance at high electric fields. However, the W_{rec} exhibits
relatively unsatisfactory behavior at medium or low electric fields. After careful
checking, the manuscript has been revised as follows:

High W_{rec} values of $\sim 10.06 \text{ J cm}^{-3}$, $\sim 8.09 \text{ J cm}^{-3}$, and $\sim 7.4 \text{ J cm}^{-3}$ have been
realized in KNN-based bulk ceramics prepared by conventional solid-state sintering at
their respective breakdown electric field of about 800 kV cm^{-1} ^{13,21,22}. While under a
moderate electric field of 500 kV cm^{-1} , these materials exhibit suboptimal W_{rec} values
of $\sim 5.6 \text{ J cm}^{-3}$, $\sim 3.8 \text{ J cm}^{-3}$, and $\sim 4.4 \text{ J cm}^{-3}$, respectively, which limit their compactness
and miniaturization for integration into electronic devices^{1,9}.

We acknowledge that the actual energy storage performance at real working
voltages is of significant importance. Nevertheless, W_{rec} serves as a fundamental
metric for evaluating material performance. A low W_{rec} may restrict the material's
use in electronic devices, regardless of the voltage applied. It fails to meet the
compactness and miniaturization demands of advanced electronic and electrical

systems. Besides, real working voltage range is much wider than household voltage
depending on the real devices, like the dielectric layer thickness in an MLCC.
Therefore, broader efforts should be made to optimize the W_{rec} under different
electric fields, rather than focusing solely on a specific or low electric field [*Energy*
*Environ. Sci.* **2020**, 13, 2938; *Energy Storage Mater.* **2020**, 30, 392; *J. Mater. Chem. A*
**2020**, 8, 683].

4. Your questions: What is the distinct role of the second compound and point defect
in energy storage behavior?

**Response:** Thank you for your question. According to the feedback, we have added
the relevant discussions as follows: **Secondary compound:** When a second
compound is incorporated into a matrix compound, it forms a homogeneous solid
solution with the 2nd compound level below 0.2. This means that the two compounds
are mixed at the atomic level, creating a uniform distribution throughout the
material. Certain secondary compounds (e.g., SNZ) can improve the overall energy
properties and stability by modulating the structure of the primary material. This
includes, but is not limited to: refining grain size and domains, introducing
heterogeneous structures, and inducing highly symmetric phase configurations. **Point**
**defects:** The defects (mainly refers to A-site vacancies in current system) induced by
introducing aliovalent cations such as Sr Nd, Zr replacing K, Na, Nb, are conducive to
increasing the ϵ_r (Fig. 3d) and thus generating a high P_{max} [*J. Mater. Chem. C* **2019**, 7,
4999-5008; *J. Mater. Chem. A* **2018**, 6, 17896-17904; *J. Mater. Chem. A* **2021**, 9,
18026-18085]. In addition, the formation of dipoles between $\text{Sr}^{2+}/\text{Nd}^{3+}$ and vacancies

at non-equilibrium positions creates a built-in internal field, which enhances the
relaxor behavior of ceramics and thus generating a low P_r [*Acta Mater.* **2020**, 197,
224-234].

In our study, the improved energy storage performance and stability are closely
related to the refinement of the grain size, the introduction of coexisting polar
nanoregions and the induction of a highly symmetric phase structure, and can thus
be primarily attributed to the second compound.

5. Your questions: Some papers reveal that rare-earth elements can improve the d_{33} ,
requiring large P_r after removing the field. Why does the P_r in Nd-doped KNN here
behave differently?

**Response:** Thank you for your insightful comment regarding the behavior of P_r in
Nd-doped KNN, particularly in comparison to other studies on rare-earth elements.
The characteristics of P_r presented in our manuscript differ due to several
contributing factors: 1) Different methods of chemical modification: KNN-based
piezoelectric materials are modified by rare-earth elements (e.g., La, Ce, Pr, Nd, Sm,
Gd, Dy, Yb, etc.) in the form of oxides, and piezoelectric properties can be controlled
by tailoring the defect types [*Chem. Rev.* 2015, 115, 2559–2595]. In our study, we
construct ABO_3 by combining Sr and Zr with Nd to modify the KNN materials through
a collective contribution from different cations, rather than single element.
Consequently, the mechanisms involved can vary significantly based on type of
chemical modifications. 2) Different structural origins lead to distinct $P-E$ loops:
Piezoelectric materials exhibit square $P-E$ loops and high P_r , aiming for enhanced

piezoelectric properties through low-symmetry phase structures and large
ferroelectric domains. In contrast, energy storage materials display slim P - E loops and
low P_r , focusing on property enhancement via high-symmetry phase structures and
refined nanodomains. 3) Comparison with existing literature: Wang et al. reported
that Nd-doped BiFeO₃-BaTiO₃ energy storage ceramics exhibit relaxor behavior
characterized by the absence of macrodomains, which leads to a reduction in P_r
[*Energy Environ. Sci.* **2019**, 12, 582-588].

6. Your questions: Why did the authors initially measure the energy storage
properties under a low field of 200 kV/cm?

**Response:** Thank you for your question regarding our choice to measure the energy
storage properties under a low electric field of 200 kV cm⁻¹. Our decision was based
on several key considerations: 1) This electric field has been utilized as a benchmark
for comparing various compositions. Low electric fields enable the observation of the
material's initial polarization behavior prior to reaching saturation or breakdown.
This information is crucial for comparing the energy storage capabilities of different
compositions and understanding the underlying mechanisms. 2) Assessing energy
storage performance under low electric fields is crucial for determining the
applicability of various materials. Numerous applications, including capacitors and
pulse power systems, operate within these conditions. Consequently, understanding
material performance in low electric fields is essential for industrial utilization [*Nat.*
*Commun.* **2024**, 15, 8651]. 3) Low electric field property measurements serve as a
standard for comparing various materials. In the literature, it is common to report an

electric field strength of either 100 kV cm⁻¹ or 200 kV cm⁻¹ [*Adv. Mater.* **2024**, 36,
2313285; *Chem. Eng. J.* **2022**, 435, 135065; *J. Mater. Chem. A* **2023**, 11, 7987-7994;
*Energy Storage Mater.* **2024**, 65, 103055; *Chem. Eng. J.* **2023**, 451, 138916]. This
enables a comprehensive evaluation of the effectiveness of various strategies,
including doping techniques, defect engineering, domain engineering, and grain size
optimization.

7. Your questions: Why do the polars with high dynamics exhibit enhanced relaxor
behavior? Why is higher relaxor behavior accompanied by low \$t_{0.9}\$?

**Response:** Thank you for your insightful questions. Polars with high dynamics exhibit
enhanced relaxor behavior due to several interrelated factors: 1) Polar nanoregions
(PNRs): In relaxor materials, polar nanoregions PNRs play a crucial role in the
dielectric response. PNRs characterized by low switching energy barriers and high
dynamics ensure fast polarization response to an applied electric field, and they are
essential for achieving slim *P-E* hysteresis loops, which confirm the enhanced relaxor
behavior [*Adv. Mater.* **2024**, 36, 2313285; *Inorg. Chem.* **2021**, 60, 6559-6568]. 2)
Local disorder: Relaxor materials are distinguished by their compositional or
structural disorder, which facilitates the movement of dipoles and enhances their
respond to external electric fields. The high dynamics observed in polar regions
enable rapid rearrangement of dipoles, thereby improving the overall relaxor
behavior. [*J. Am. Chem. Soc.* **2024**, 146, 460-467]. 3) Dielectric relaxor: High dynamic
polar regions facilitate faster dielectric relaxor. When an electric field is applied, the
material rapidly polarizes and depolarizes, characteristic of relaxor behavior. [*J. Am.*

*Chem. Soc.* **2024**, 146, 13467-13476; *J. Am. Chem. Soc.* **2023**, 145, 19396-19404]. 4)
Collective behavior: In dynamic materials, polar regions show collective behavior,
where one dipole's movement influences others. This enhances relaxor
characteristics [*Energy Storage Mater.* **2019**, 18, 238-245]. In summary, the high
dynamics in polar regions, characterized by fast polarization response of PNRs,
increased local disorder, faster dielectric relaxor, and collective behavior, all
contribute to enhancing the relaxor behavior in these materials.

Theoretically, the discharge speed, represented by $t_{0.9}$, indicates the time
required for the discharged energy to reach 90% of its final value. This parameter
relates to the material's response to an external electric field. In relaxor materials,
the efficient polarization and depolarization processes lead to shorter discharge
12 times, hence a lower $t_{0.9}$. However, it is crucial to emphasize that the $t_{0.9}$ is controlled
mainly by the capacitance of the ceramics and external load resistor (R) [*Science*
**2006**, 313, 334-336; *Adv. Energy Mater.* **2013**, 3, 451-456; *Nano Lett.* **2013**, 13,
1373-1379]. For instance, in the current system, $t_{0.9}$ increases with enhanced relaxor
behavior under an electric field of 100 kV cm⁻¹ (**Fig. R7**), mainly due to improved
capacitance. Therefore, the correlation between enhanced relaxor behavior and a
lower $t_{0.9}$ may be valid under certain specific conditions. In fact, It is hard to
accurately assess stronger relaxor.

Fig. R7 Comparison of $t_{0.9}$ for (1-x)KNN-xSNZ ceramics (pure KNN, x = 0.03, x = 0.15).

8. Your questions: How do the authors justify the assumption: “The origin is inferred to be the substitution of Nd^{3+} at the A-site, leading to the generation of vacancies and expansion of the oxygen octahedron”?

Response: Thank you for your insightful comments regarding the assumption of Nd^{3+} substitution at the A-site and its effects on the crystal structure. 1) The necessity for charge neutrality within the crystal lattice is indeed critical. The substitution of Nd^{3+} , which has a different valence state compared to the original A-site cation, necessitates the formation of vacancies. This can be expressed by Kroge-Vink notation [*J. Eur. Ceram. Soc.* **2017**, 37, 1429-1436; *Nat. Mater.* **2014**, 13, 31-35; *Energy Storage Mater.* **2020**, 30, 392-400]:

Besides, we sincerely apologize for neglecting Sr^{2+} , which is similar to Nd^{3+} :

2) We acknowledge the need for clarification regarding the expansion of the oxygen octahedron. The increase in the ratio of lattice parameters (c/a) suggests that the

1 oxygen octahedron may be expanding along the c-axis [*J. Mater. Chem. C* **2018**, 6,
8528-8537; *J. Materiomics* **2024**, 10, 179-189]. The orthorhombic lattice constants
(a_o and c_o) and tetragonal lattice constants (a_T and c_T) can be determined through
refinement analyses. For instance, pure KNN (**Fig. R8**) ceramic has a O structure with
a typical ferroelectric/piezoelectric behavior and lattice constants of $a_o = 7.998 \text{ \AA}$
and $c_o = 7.989 \text{ \AA}$ with an orthogonality of $c_o/a_o = 0.999$. In contrast,
0.85KNN-0.15SNZ ceramic (Fig. 3a) reveals the coexistence of the O and T phases,
with weight fractions of 26.14% and 73.86%, respectively, and $c_o/a_o = 1.418 \text{ \AA}$, c_T/a_T
= 0.998 \AA . 3) However, It is crucial to emphasize that the structure of (1-x)KNN-xSNZ
cannot be merely regarded as a combination of O and T phases. The observed
changes in lattice parameters only indicate a distortion of the oxygen octahedra. 4)
Ionic radii: K^+ (1.64 \AA , CN = 12) and Na^+ (1.39 \AA^+ , CN = 12) can be effectively
substituted by Sr^{2+} (1.44 \AA , CN = 12) and Nd^{3+} (1.27 \AA , CN = 12), while Zr^{4+} (0.72 \AA , CN
= 6) can replace Nb^{5+} (0.64 \AA , CN = 6). The substitution of the smaller ionic radius and
the shorter bond length, as determined from the Rietveld refinement results, further
indicates that the oxygen octahedron may not have experienced expansion.

22 **Fig. R8** Rietveld refinement XRD pattern of pure KNN ceramic.

After careful consideration, we have revised the manuscript as follows
(reference 49 has also been added):

The origin of a built-in internal field is inferred to be the substitution of Sr²⁺
and Nd³⁺ at the A-site, leading to the generation of vacancies and distortion of the
oxygen octahedron. This results in a more facile displacement of ions at the B-site
and enhanced polarization. In addition, the formation of dipoles between Sr²⁺/Nd³⁺
and vacancies at non-equilibrium positions creates a built-in internal field, which
enhances the relaxor behavior of ceramics⁴⁹.

9. Your questions: Why does the C phase in Fig. 5b display local polarization? I am
concerned that the evidence of C/T/O coexistence is insufficient.

**Response:** Thanks so much for your insightful comments regarding the local
polarization observed in the C phase and the concerns about the evidence for C/T/O
coexistence. It is accurate to state that the C phase is typically symmetric and does
not display local polarization due to the balanced distribution of charges. Fig. 5b
presents the corresponding local polarization configuration of Fig. 5a derived from
the 2D Gaussian peak fitting and quantification. Consistent with the literature
[*Science* **2019**, 365, 578-582; *Nat. Commun.* **2022**, 13, 3089; *Adv. Mater.* **2022**, 34,
2205787; *Energy Environ. Sci.* **2023**, 16, 4511-452], the C phase in our manuscript can
also be observed by the arrows with almost no polarization magnitude. We
delineated the regions with pink dashed lines, as illustrated in **Fig. R9**.

Fig. R9 Quantified local polarization configuration corresponding to Fig. 5a.

Theoretical context: In KNN materials, O phase is typically regarded as the phase with the lowest energy. However, doping with paraelectric phase particles such as SNZ reduces the Landau energy barrier of the material system, making polarization easier to switch, which in turn leads to the easier formation of T phase and C phase. **Experimental evidence:** In addition to confirmation through aberration-corrected scanning transmission electron microscopy (Fig. 5a, b), Fig. 3a illustrates the Rietveld refinement results for the 0.85KNN-0.15SNZ ceramic. It reveals the coexistence of the O and T phases, which is a characteristic feature of a typical pseudo-cubic structure. The exceptionally broad and flat dielectric peaks (Fig. 3c), combined with the consistent characteristic peaks observed through temperature-dependent powder XRD techniques, suggest that there are no changes in phase structure (Fig. 3b) within the investigated temperature range of 25-600 °C. This further supports the coexistence of the O, T, and C phases. Furthermore, the selected area electron diffraction patterns taken along [001]_c and [110]_c directions indicate that the sample exhibits an average cubic structure, as illustrated in Fig. 3e

and Fig. 3f, respectively. The diffused and weak spots as indicated by white arrows
confirm the existence of orthorhombic structure “unit cell” which is orientated along
the cubic [110]c in the x-direction, the [1-10]c in the y-direction and the [001]c in the
z-direction. Additionally, XRD patterns (Fig. S4), Raman spectra (Fig. S5), and
simulations (Fig. S8) provide further support for the coexistence of C/T/O phases,
which we have discussed in detail in our original manuscript.

10. Your questions: Why doesn't the phase field simulation show the coexistence of
C/T/O states? I believe it is difficult to induce the C phase in this composition.

**Response:** Thanks so much for your reminding and question. We understand your
concern about our phase field simulations. In ferroelectric materials, C phase is
regarded as a symmetric crystal structure, which means there is no polarization
caused by the displacement of positive and negative charges. In our phase-field
model, the order parameter is the polarization of unit cells, so we can only
differentiate the C phase (paraelectric phase) and the ferroelectric phase based on
the domain size and the magnitude of polarization. As we only take the direction of
polarization into consideration, we ignored the non-ferroelectric phase when plotting
the domain structure. Nevertheless, we manually marked the O/T/C phase in the
vector plot of the supporting material (see fig. S8).

Additionally, in KNN materials, the O phase is generally considered the lowest
energy phase. However, doping with paraelectric phase particles, such as SNZ,
reduces the Landau energy barrier, facilitating easier polarization switching. This
process contributes to the formation of both T and C phases. Furthermore,

numerous experimental studies have confirmed the existence of the C phase, as
discussed in our response to the previous question.

11. Your questions: Why is the discussion part the conclusion?

**Response:** Thank you for your insightful question about our manuscript's structure,
especially the discussion section. In Nature Communications, there is usually no
separate "Conclusion" section; instead, conclusion content is integrated into the
"Discussion" or "Results" sections. This integrated approach simplifies the
manuscript's structure, improves clarity, highlights the research's significance, and
adheres to Nature Communications' formatting guidelines. Ultimately, it enables a
more coherent and impactful presentation of the findings.

We really appreciate the time and effort that the reviewers have invested in
evaluating our manuscript entitled “Excellent energy storage properties in lead-free
ferroelectric ceramics via heterogeneous structure design” (NCOMMS-24-49092A).
We have carefully read the reviewers’ feedback, all the comments and suggestions
are important to improve the quality of our manuscript. We have revised our
manuscript accordingly, and the point-by-point response to the comments are
enclosed in this response letter. In order to make the changes easily readable, we
highlighted our revisions in the revised manuscript.

**Reviewer #1**

**General comments:**

The authors have addressed all the comments thoroughly and made meaningful
revisions that enhance the manuscript's clarity and scientific rigor. I am a big fan of
Table R1 to distinguish the contributions of each mechanism. Hence, I recommend
accepting the revised manuscript after ensuring all changes are correctly integrated
into the main text and supplementary materials.

We sincerely appreciate your valuable feedback and suggestions on our
manuscript. We have evaluated the accuracy of Table R1 and other changes based on
your suggestions, and incorporated them into our main text and supplementary
materials (All changes in the revised manuscript are highlighted in blue font, while
the revised supplementary materials are presented without highlights due to
formatting requirements).

**Reviewer #3**

**General comments:**

We appreciate the authors' careful responses and revisions to the manuscript.
However, there are still two issues that need to be addressed before the manuscript
can be considered for publication.

We sincerely appreciated for your valuable feedbacks and detailed comments
on our manuscript. According to your comments, the manuscript has been revised as
follows, which I have replied to point by point below (All changes in the revised
manuscript are highlighted in **green font**, while the revised supplementary materials
are presented without highlights due to formatting requirements).

1. Your questions: The authors have listed several papers reporting high energy
storage densities, but these values are lower than the 14 J/cm³ reported in the
current work. However, recent literature appears to include higher energy storage
densities, such as 15.9 J/cm³ (Nature Communications, 2024, 15:7338) and 15.48
J/cm³ (Nature Communications, 2024, 15:6754). These references should be included
in the manuscript, particularly in the introduction section and in Figure 2 (a), (b), and
(h).

**Response:** Thank you for your questions. We agree that the manuscript should
include recent literature on higher energy storage densities, particularly in the
introduction and Figure 2h. Figure 2a and Figure 2b indicate that the current work
represents a significant breakthrough in KNN-based ceramics, leading to the
exclusion of other lead-free systems from our summary. The energy storage densities

and discharge energy densities from the references (*Nat. Commun.* **15**, 6754, 2024;
and *Nat. Commun.* **15**, 7338, 2024) are included in Supplementary Tables S1 and S3,
and Figure 2h, respectively. Additionally, we have modified the introduction section,
the description of Figure 2h, and the references in our revised manuscript
accordingly.

2. Your questions: The authors describe three key factors contributing to the
enhanced energy storage performance: (1) dense microstructure, (2) relaxor behavior,
and (3) minimal impurities. However, the manuscript primarily discusses small grains
and certain nanodomains, with supporting evidence for grain size provided in the
supplementary materials. It remains unclear how the small grain size relates to the
proposed heterogeneous strategy. This connection should be clarified in the
manuscript.

**Response:** Thanks for your question. According to this suggestion, we have added
the relevant discussions as follows:

When explaining energy storage properties of lead-free ceramics, both intrinsic
and extrinsic contributions are considered. The intrinsic contribution is related to
their lattice arrangement (e.g., lattice distortion), while the extrinsic contribution
mainly originates from their microstructure, including ferroelectric domains, grain
size, density, and porosity. When a ceramic is crushed, countless grains are obtained.
These grains are comprised of numerous complicated ferroelectric domains, which
are attributed to the lattice arrangement. Therefore, the domain structure acts as a
bridge connecting the lattice alignment with macro grains (or ceramics) [*J. Mater.*

*Chem. A*, **8**, 10026-10073, 2020].

The optimized properties in RFEs originate from their weakly intercoupled
nanodomains or PNRs that induce lower energy barriers for polarization switching,
instead of the strongly intercoupled micrometer-size domains in typical FEs. The
generally accepted approach to realizing RFEs is to disrupt long-range FE order into
nanodomains [*Science* **365**, 578-582, 2019]. Due to the confinement imposed by
grain boundary conditions, the size of domains decreases as grain size decreases [*Adv.*
*Mater.* **36**, 2403400, 2024]. In other words, the refined grain boundaries can
effectively hinder domain growth, which is an important factor for achieving PNRs.
Additionally, the domain size decreases as the B-site cation displacement vectors
become more disordered and vice versa. Therefore, the heterogeneous is
constructed via a multi-scale optimization strategy from grain scale, to domain scale
to atomic scale. To clarify the relationship between small grain size and the proposed
heterogeneous strategy, we have included the schematic diagram in the
supplementary materials as Supplementary Fig. S1 and incorporated this connection
into our revised manuscript.